# The Expanded Othello AI Arena: Evaluating Intelligent Systems Through Constrained Adaptation to Unseen Conditions

**Byunghwa Yoo**                                                      *blakeyoo@gm.gist.ac.kr*
*Department of AI Convergence*
*Gwangju Institute of Science and Technology*

**Sundong Kim**[*]                                                    *sundong@gist.ac.kr*
*Department of AI Convergence*
*Gwangju Institute of Science and Technology*

**Kyung-Joong Kim**                                                   *kjkim@gist.ac.kr*
*Department of AI Convergence*
*Gwangju Institute of Science and Technology*

**Reviewed on OpenReview:** *https://openreview.net/forum?id=WXKQtqPC2d*

## Abstract

The ability to rapidly adapt to environmental changes is a core requirement for Artificial General Intelligence (AGI), yet most AI benchmarks evaluate performance in static environments. We present the Expanded Othello AI Arena, a benchmark designed to measure Skill-Acquisition Efficiency: the rate at which agents discover latent objectives and converge to effective strategies within a limited interaction budget. The Arena formalizes a spectrum of 56 environments using a parametric framework $\mathcal{E} = (\mathcal{L}, \mathcal{C})$, where $\mathcal{L}$ defines Othello board geometries and $\mathcal{C}$ represents latent winning conditions via a disc-ratio threshold $K$. This parameterization requires agents to decipher terminal rules through direct interaction while adapting against an opponent in a zero-sum setting; in narrow regimes, agents must precisely control their terminal occupancy under the hidden threshold, while terminal outcomes in which neither player satisfies the admissible interval are treated as draws. Unlike traditional evaluation, the Arena imposes a strict interaction budget to prioritize sample efficiency over asymptotic optimization. We establish the benchmark's utility through a neuroevolutionary adaptive-Minimax baseline that utilizes meta-learned spatial priors and adaptive weighting. Our empirical analysis reveals that while this baseline achieves competitive performance in standard and inverse regimes, it struggles in narrow-interval regimes that demand precise terminal control under latent objectives. Released as an extensible Python-based research toolkit, the Arena provides a standardized platform for exploring research directions, including test-time learning, in-context learning, and world models. Our project page is available at: https://expanded-othello.vercel.app/

## 1 Introduction

The capacity for rapid strategic adaptation to novel environments has been one of the primary goals for Artificial General Intelligence (AGI) (Chollet, 2019). In recent years, specialized artificial agents have achieved remarkable milestones in various domains. For instance, reinforcement learning (RL) agents have reached superhuman proficiency in high-complexity board games such as Go, Chess, and Shogi (Silver et al., 2016; 2017; 2018), and large language models (LLMs) have demonstrated their reasoning abilities in text-based games (Yang et al., 2024; Zhuo & Murata, 2024; Hu et al., 2025; Phan et al., 2025) and complex, long-horizon

---

[*]Corresponding author.

video games such as Minecraft (Wang et al., 2024). However, these successes are predominantly anchored in knowledge-saturated environments, where the agents benefit from either an extensive amount of training interactions or decide actions zero-shot, purely based on prior knowledge.

To mitigate such properties of traditional environments, various studies have proposed environments that incorporate the variation concept, such as ProcGen (Cobbe et al., 2020) and MiniHack (Samvelyan et al., 2021). These benchmarks challenge agents to generalize across procedurally generated levels, requiring agents to learn to adapt to novel environments. However, the novelty or the variation introduced by these environments is limited. For instance, ProcGen focuses on the variation of visualizations while the set of games remains the same, and MiniHack offers varying level layouts and tasks within a fixed rule-set. Nonetheless, while these environments serve as a great testbed for traditional meta-learning, they typically do not deal with changes to core game dynamics, such as reversed winning conditions. Furthermore, most existing benchmarks evaluate adaptation through single-player interactions with the environment alone, thereby overlooking the necessity of adaptation under competitive interactions, where an agent must respond to an opponent whose behavior co-determines the outcome. This oversight limits the evaluation of an agent's ability to identify hidden reward structures and objective functions from sparse terminal signals, which is a critical facet of general intelligence that involves not only understanding environmental structures but also identifying the fundamental objectives of the interaction itself.

Therefore, we introduce the Expanded Othello AI Arena, a benchmark designed to move beyond structural or visual perturbations by incorporating objective-level novelty within a competitive framework. Unlike existing platforms, the Arena compels agents to infer terminal game rules and adapt their strategic behavior within a strictly constrained interaction budget while contending with an opponent, thereby shifting the focus toward skill-acquisition efficiency under latent objectives in adversarial play. Consider an agent that dominates a game with 44 discs against 20, yet unexpectedly fails to win because the hidden rule penalizes excessive occupation; can an intelligent system identify such a non-obvious objective from sparse terminal signals within a limited number of trials? The Expanded Othello AI Arena is designed to answer this question.

To systematically evaluate these capabilities, we formalize the environment space of the Expanded Othello AI Arena as a parametric framework $\mathcal{E} = \mathcal{L} \times \mathcal{C}$, defined as the Cartesian product of the layout space $\mathcal{L}$ and the condition space $\mathcal{C}$. Each environment instance $E \in \mathcal{E}$ is represented as a tuple $(L, C)$, where $L \in \mathcal{L}$ specifies the board geometry and obstacle placements fully observable to the agent. In contrast, $C \in \mathcal{C}$ denotes the hidden winning conditions, parameterized via a disc-ratio threshold $K$, which must be inferred solely through a limited number of interactions with the environment $E$. This $K$-parameterization generalizes standard, inverse, and narrow-interval victory states. Furthermore, since Othello works as a 1-vs-1 zero-sum setup, successful adaptation demands not only identifying $K$ but also adapting strategic behavior to the opponent. Consequently, the Arena serves as a controlled micro-world in which agents must adapt their strategy while uncovering the latent rules governing the task under interaction with an adversary.

By imposing a strict interaction budget, the Arena shifts the evaluation focus toward the velocity of strategic convergence rather than asymptotic optimization. This constraint exposes a critical efficiency gap that current optimization-heavy agents often fail to bridge, since agents must adapt to novel geometries ($L$) and latent winning conditions ($C$) from a restricted stream of experience. It therefore provides a compact testbed for studying how quickly agents adjust to unfamiliar game variants (Chollet, 2019; Reed et al., 2022).

To demonstrate the practical utility and discriminative power of the Arena, we provide a foundational baseline utilizing a neuroevolutionary adaptive-Minimax architecture. This hybrid system leverages meta-learned spatial priors combined with a dynamic utility-weighting mechanism that adapts to latent winning conditions ($\mathcal{C}$). Our empirical analysis reveals that while this baseline achieves significantly higher sample efficiency than traditional reinforcement learning agents within the limited interaction budget, a substantial performance ceiling remains in the most complex environment variants. By releasing the Arena as a modular, local Python-based research toolkit, we aim to provide a standardized testbed for investigating how intelligent systems can infer latent objectives from sparse terminal signals and align their strategies accordingly—capabilities that are central to general intelligence yet remain underexplored in existing benchmarks.

In summary, the primary contributions of this work are:

- **Formalization of a Parametric Environment Space**: We introduce a framework for evaluating adaptation under latent objectives and varying geometries by formalizing the environment space as $\mathcal{E} = \mathcal{L} \times \mathcal{C}$, decoupling observable board geometries ($L$) from latent winning conditions ($C$).

- **Adaptation under Latent Objectives in Adversarial Play**: We propose a benchmark where adaptation requires the simultaneous deciphering of latent rules and strategic adjustment against an opponent in a zero-sum context, under sparse terminal feedback and a limited interaction budget.

- **The Expanded Othello AI Arena Toolkit**: We release a modular, Python-based research toolkit designed to quantify Skill-Acquisition Efficiency under strict interaction budgets and analyze the velocity of strategic convergence.

- **Empirical Baseline and Efficiency Analysis**: We establish the benchmark's utility through a neuroevolutionary baseline, identifying a substantial efficiency gap under limited interaction budgets.

## 2 The Expanded Othello AI Arena Environment

To evaluate an agent's capacity for rapid strategic adaptation, we formalize a parametric environment space $\mathcal{E} = \mathcal{L} \times \mathcal{C}$ built upon the core mechanics of Othello. The Arena is designed as a rigorous proving ground for Skill-Acquisition Efficiency, specifically evaluating an intelligent system $I$ — a combination of a learning algorithm and a corresponding model — on its ability to adapt to novel rules and adversarial challenges.

### 2.1 Adaptation Formalization

The adaptation task in the Arena is defined as the process by which a system $I$ interacts with a specific environment instance $E = (L, C)$ to adapt its policy $\pi$ into an environment-specific strategy $\pi_E$. This transition is strictly limited by an interaction budget of $N_{games} = 2,000$ episodes. We formalize this transformation as:

$$I(E) : \pi \xrightarrow{N_{games}} \pi_E \tag{1}$$

In this framework, the primary subject of evaluation is the efficiency of system $I$. The goal is to measure how effectively the system's underlying algorithm can transition from its general baseline state $\pi$ to an optimized state $\pi_E$ by deciphering terminal signals and navigating the strategic uncertainty of the environment within the allowed scarce interaction opportunities.

This formulation shifts the evaluation from asymptotic performance to the velocity of strategic convergence, focusing on how an intelligent system $I$ identifies environmental conditions with respect to the opponent's unpredictable actions to produce a functional, task-specific policy $\pi_E$.

### 2.2 Game Dynamics and Substrate Rationale

Othello (Reversi) is a classic zero-sum, perfect-information board game characterized by the strategic reversal of opponent discs through spatial sandwiching mechanics. We select Othello as the foundational substrate for our benchmark due to its unique combination of simple local rules and high global state-change dynamics. Unlike games with static piece values, Othello involves frequent state reversals (flipping), making it an ideal environment where the geometric layout of the board directly affects core strategies. Furthermore, the game's terminal state naturally maps to a continuous disc occupancy ratio $\rho \in [0, 1]$, providing a parameterizable winning condition variation.

The Expanded Othello framework maintains the two fundamental mechanics of traditional Othello as its invariant core:

**Flipping Mechanism**: A player places a disc on an empty cell to sandwich one or more opponent discs between the new disc and an existing disc of their own, along any of the eight possible directions.

**Legal Moves and Termination**: A move is only legal if it flips at least one opponent disc. The game ends when neither player can make a legal move.

While these core dynamics remain constant, the Expanded Othello introduces novelty and complexity by perturbing the spatial layout and the terminal victory conditions, forcing the agent to adapt and discover the dynamics of the novel environment it is confronted with.

## 2.3 Formalization of the Environment

We define the environment space $\mathcal{E}$ as the Cartesian product of a layout space $\mathcal{L}$ and a condition space $\mathcal{C}$:

$$\mathcal{E} = \mathcal{L} \times \mathcal{C} \tag{2}$$

Each environmental instance $E \in \mathcal{E}$ is represented as a tuple $E = (L, C)$. A critical feature of the Expanded Othello is the asymmetry of information, where the spatial layout $L$ is fully observable to the agent, while the victory condition $C$ remains latent.

### 2.3.1 The Observable State and the Layout

In the Expanded Othello, the observable state at turn $t$ is defined by the spatial configuration of discs and obstacles on the board. This state is represented as a grid $G_t \in \{-1, 0, 1, 2\}^{W \times H}$, where the cell values are assigned as follows: 1 and $-1$ denote the discs of the agent and the opponent, respectively; 0 represents an empty playable cell; and 2 signifies an immutable obstacle. Obstacles are permanently unplayable and obstruct the flipping logic; specifically, discs cannot be sandwiched across an obstacle.

The layout $L$ is formally defined as the initial board configuration $G_0$, and thus remains static throughout the game. While the positions of discs evolve, the distribution of obstacles is fixed upon environment initialization, introducing significant spatial diversity. By incorporating irregular board boundaries and non-interactive obstacles, the Expanded Othello effectively disrupts the traditional heuristics of standard Othello, as such methods are topology-dependent on the classical Othello (Rose, 2005; Yoshioka et al., 1999; Lucas, 2008; Jaśkowski, 2014; Szubert et al., 2013; Buro, 1995). This forces agents to generalize spatial value-mapping across diverse and novel topologies.

### 2.3.2 The Interval-Based Victory Condition

While the layout $L$ provides the spatial context, the condition space $\mathcal{C}$ defines the strategic constraints of the environment. Unlike traditional games where dominance strictly correlates with victory, Expanded Othello parameterizes success through a latent threshold $K \in \mathbb{R} \setminus \{0.5\}$, introducing a relative victory logic based on the occupancy ratio $\rho \in [0, 1]$, which is defined as follows for each player's number of discs on the board $n$:

$$\rho = \frac{n_{agent}}{n_{agent} + n_{opponent}} \tag{3}$$

The threshold $K$ induces an *admissible interval* $\mathcal{I}(K)$ that determines the winning occupancy range:

$$\mathcal{I}(K) = \big( \min(K,\ 0.5),\ \max(K,\ 0.5) \big) \tag{4}$$

The terminal outcome is then determined by a single unified rule: Player $i$ wins if and only if their terminal occupancy satisfies $\rho_i \in \mathcal{I}(K)$. If neither player's terminal occupancy falls within $\mathcal{I}(K)$, the outcome is a draw. Note that since $\rho_i + \rho_j = 1$, at most one player can have $\rho \in \mathcal{I}(K)$ at any terminal state, so simultaneous victory is impossible.

By varying $K$, this formulation recovers several distinct Othello variants within a single parameter:

**Standard (Positive) Othello ($K > 1.0$):** By setting $K > 1.0$, the admissible interval effectively becomes $(0.5, 1.0]$ since $\rho \in [0, 1]$. This recovers the traditional majority-win rule, where the objective is to simply maximize disc occupancy.

**Narrow-Majority Othello ($0.5 < K \leq 1.0$):** The admissible interval becomes $(0.5, K)$, requiring the agent to seek a majority ($\rho > 0.5$) while keeping its terminal occupancy strictly below $K$.

**Inverse (Negative) Othello ($K < 0$)**: By setting $K < 0.0$, the admissible interval effectively becomes $(0, 0.5)$. This recovers the simple minority-win rule, where the objective is to minimize disc occupancy.

**Narrow-Minority Othello ($0 \leq K < 0.5$)**: The admissible interval becomes $(K, 0.5)$, requiring the agent to maintain a minority ratio while staying strictly above $K$.

The narrow regimes introduce a distinct requirement of terminal occupancy control. When the admissible interval is very wide (i.e., $K \simeq 1.0$ or $K \simeq 0.0$), the agent can often rely on broad majority or minority-seeking behavior. In contrast, as the interval becomes narrower, successful play depends not only on discovering whether the environment rewards majority or minority control, but also on regulating the final occupancy to remain within a limited target range. Outcomes that overshoot this range are not counted as victories, even if they would correspond to dominant play under standard Othello. Furthermore, in extreme cases such as $K = 0.52$, the admissible interval becomes nearly impossible to satisfy — requiring, on a fully occupied standard $8 \times 8$ board, an occupancy of exactly 33 discs ($\rho \in (0.5, 0.52) \Rightarrow \#\text{discs} \in (32, 33.28)$). In such scenarios, the strategic priority shifts from simple occupancy maximization toward precise control of the terminal ratio so that the draw condition can be easily met.

Consequently, the Expanded Othello environment is transformed from a stationary optimization task into a test of online system identification and strategic adaptation under adversarial interaction. Because the winning condition $C$ remains latent, the agent must not only decipher the hidden success criteria from sparse terminal feedback but also adjust its play in response to an opponent whose actions co-determine the final occupancy ratio.

For instance, consider a majority-constraint regime with $K = 0.8$, where $\mathcal{I}(K) = (0.5, 0.8)$. Suppose at the terminal state, two players $i$ and $j$ have disc occupancy ratios $\rho_i = 0.25$ and $\rho_j = 0.75$. Player $j$ wins, as $\rho_j \in (0.5, 0.8)$. Now suppose instead that $\rho_j = 0.85$; despite dominating the board, player $j$ exceeds the threshold $K = 0.8$, and neither player falls within $\mathcal{I}(K)$, resulting in a draw. This illustrates a key property of the Arena: a player who has already achieved majority control must avoid *over-domination* that would push the terminal ratio beyond $K$, while a losing player may benefit from forcing the opponent's occupancy above the threshold rather than attempting to reclaim majority. Note that in practice, Othello's flipping mechanics make intermediate occupancy ratios a poor predictor of terminal outcomes, further compounding the challenge of precise occupancy control.

## 2.4 Interaction Protocol

The interaction between an agent and the environment is designed to isolate the challenge of strategic adaptation. While the mathematical properties of the state $G_t$ and the terminal outcome conditions are established in Section 2.3, this section defines the functional interface and the specific flow of information governing the interaction loop.

**Action Space ($\mathcal{A}$)**: At any given turn $t$, the set of legal actions $\mathcal{A}_t$ is determined by the current board state $G_t$. An action $a_t \in \mathcal{A}_t$ is a coordinate $(x, y)$ within the grid boundaries $W \times H$ where a disc can be placed. A move is legal if both conditions are met: First, the target cell is empty ($G_t(x, y) = 0$). Second, the placement results in the flipping of at least one opponent disc, while not being obstructed by immutable obstacles. If $\mathcal{A}_t$ is empty, the agent is forced to pass its turn. The interaction continues until the termination condition, where neither player has a legal move. Note that the full-disc board also results in termination, as neither player has space to play their disc.

**Asymmetric Observability**: The protocol enforces a strict boundary between observable spatial data and latent objective data so that adaptation must proceed from board evolution and terminal outcomes alone:

- **Observable Input**: The agent receives the full grid $G_t$ as input, providing complete information regarding the layout $L$ with obstacle distributions and the current positions of all discs. Alongside the spatial observations, the set of legal action positions $\mathcal{A}_t$ is also given to the agent.

- **Latent Constraint**: The winning threshold $K$ and the resulting victory regime are never explicitly provided to the agent. The agent should operate under the assumption that the objective function is unknown and may vary between environment instances.

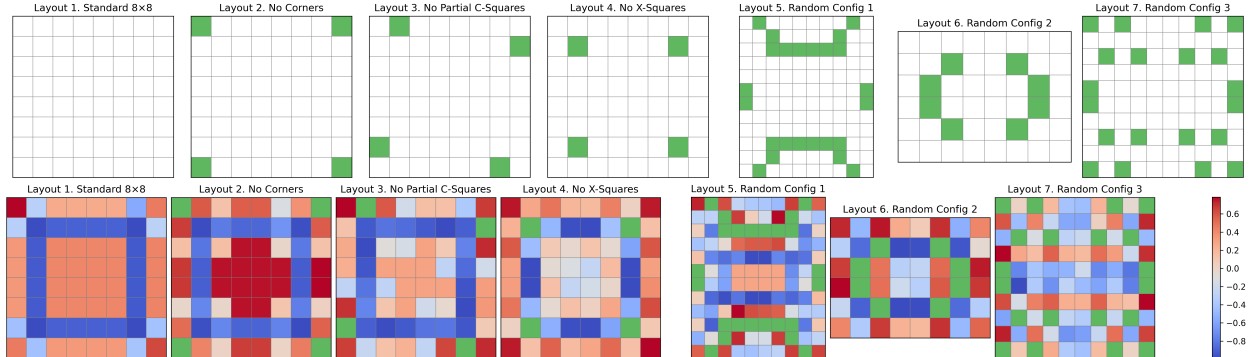

Figure 1: (Top) Layouts of the proposed standard benchmarking environments. (Bottom) The positional value tables generated via neuroevolutionary positional heuristics (PosNet) in standard condition ($K > 1.0$), where higher values are marked as red and lower values are marked as blue. Obstacles are marked as green.

**Environmental Access**: Access to an environment instance is limited to full game episodes under the standard interaction loop, limited by the $N_{games}$ budget. That is, information about the environment may be acquired only through actually playing games from the initial state to termination. Self-play is allowed. However, no additional simulator access is permitted beyond such sequential interaction: the protocol does not allow forward queries, branching rollouts, counterfactual state evaluations, or the use of a separately specified external opponent policy.

**Feedback Mechanism**: To prevent the agent from relying on dense rewards that may convey cues about the latent condition of the environment, the Expanded Othello provides a sparse terminal reward system. Throughout the episode, all intermediate rewards are given as zero, i.e., no intermediate evaluation signal is returned. Only at the terminal state $T$ does the environment return the trinary reward $R_i \in \{+1, 0, -1\}$, where $R_i = +1$ if player $i$ wins under the terminal outcome conditions in Section 2.3.2, $R_i = -1$ if player $j$ wins, and $R_i = 0$ if the terminal outcome is a draw.

This protocol ensures that any performance gain is a direct result of the agent's ability to correlate its actions and the resulting spatial configurations with the sparse terminal signals, effectively measuring its efficiency in deciphering latent strategic constraints.

## 3 Standardized Benchmarking Setup

To rigorously evaluate the adaptation capabilities of AI agents, we establish a standardized benchmarking framework. This setup is designed to move beyond static optimization, focusing instead on the efficiency gap — the disparity between rapid system identification and asymptotic convergence under severe interaction budget constraints.

### 3.1 Benchmarking Instances

The evaluation suite consists of 56 independent environmental instances, derived from a systematic $7 \times 8$ cross-product of spatial and objective variables. This ensemble ensures a comprehensive assessment across diverse topological and objective-level challenges.

**Layout Diversity ($\mathcal{L}$)**: We employ 7 distinct board layouts ($L$) as illustrated in Figure 1 (top). These include: (1) a standard $8 \times 8$ grid without obstacles; (2) obstacles on corners; (3) obstacles on partial C-squares (horizontal and diagonal neighbors of corners); (4) obstacles on X-squares (diagonal neighbors of corners); and (5-7) randomly generated obstacle distributions with varying board sizes. These irregular boundaries disrupt traditional topology-dependent heuristics, forcing agents to generalize spatial value-mapping.

**Latent Strategic Objectives ($\mathcal{C}$)**: For each layout, we apply 8 distinct victory conditions ($C$). These include the standard Othello ($K > 1.0$), the inverse Othello ($K < 0$), and various narrow-interval regimes

($K \in \{0.8, 0.6, 0.4, 0.2\}$). Additionally, we include time-constrained variants in which the game terminates after 10 turns (i.e., 10 moves per player), and the outcome is determined by the terminal occupancy ratio under $K > 1.0$ and $K < 0$ conditions. These variants evaluate whether agents can adapt their strategies under shorter match horizons.

## 3.2 Evaluation Constraints

To focus on sample efficiency rather than asymptotic convergence, we enforce strict operational limits:

**Interaction Budget**: Each agent is allocated a total of 2,000 episodes per environment ($N_{games} = 2000$). Within this limit, the agent must perform both latent rule discovery and strategy adaptation.

**Knowledge Reset**: Each of the 56 sessions is independent. All learned parameters or inferred knowledge about the evaluation environment are reset upon transitioning to a new environment, ensuring the measurement of immediate online adaptation.

**Prior Knowledge**: As the benchmark evaluates online adaptation to latent objectives, the winning condition $C$ of each evaluation instance must be discovered solely through the 2,000-game interaction budget. However, spatial prior knowledge about board topologies ($L$) may be acquired in advance, as obstacle placements are a standard variation one could assume—for example, by meta-learning positional heuristics across a distribution of layouts, as demonstrated by our PosNet baseline (Section 4.2). Similarly, the use of general-purpose foundation models (e.g., LLMs) that encode broad reasoning capabilities is permitted, provided that such knowledge is not specific to the 56 evaluation instances.

## 4 Baselines

To evaluate skill-acquisition efficiency in the Expanded Othello AI Arena, we establish a set of baselines. The defining challenge of this benchmark is the latent nature of the victory condition $C$. Because the objective is unknown, defining a computable hand-crafted fitness function aligned with the latent objective or a dense reward signal is not straightforward. Consequently, we focus on survival-only evolution as the primary mechanism for both prior knowledge acquisition and online adaptation.

### 4.1 Survival-Only Evolution Framework

The core of our approach is the evolutionary framework driven solely by terminal match outcomes (Victory/Draw/Defeat) following the conditions of Section 2.3.2. In this pure-survival scheme, agents do not receive intermediate rewards or gradient information. Instead, they undergo a comparative-selection process: individuals within a population compete in matches, where the parameters of winners are duplicated and mutated while those of losers are discarded. This ensures that adaptation is driven strictly by survival in the face of latent environmental rules, providing a robust path to strategic discovery when explicit fitness functions are unavailable. The detailed procedure for survival-only evolution is in Appendix B.

### 4.2 Meta-Learning of Spatial Priors

The historical strategies for the classical Othello have been heavily dependent on the topology of the board (Rose, 2005). For instance, corners have been considered as the most important cells, as the discs on the corners cannot be flipped. It is widely accepted that the cells in the grid of the board do not share equal values. Consequently, the early artificial agents for Othello adopted the idea of a positional value table (Yoshioka et al., 1999), a static table that scores each cell's value.

Expanding upon this concept, we seek a universal value table generator conditioned by the layout $L$. We propose the Positional Network (PosNet), a fully-convolutional network designed to generate these positional value tables as a foundation for spatial reasoning across diverse geometries. Crucially, as the winning condition $C$ is latent and an explicit fitness function cannot be defined, PosNet is trained via survival-only neuroevolution. It meta-learns generalized spatial expertise across a wide distribution of 1,000 randomly generated layouts with $C$ conditioned as $K > 1.0$. In our benchmarking framework, PosNet serves as a non-adaptive baseline

(Positional Agent) by remaining static during the interaction phase, providing a rigorous measure of the performance limits of pure spatial expertise in the absence of objective-level adaptation. The detailed algorithm for PosNet is described in Appendix C.

### 4.3  Adaptive Minimax Agent

Building upon the architectural principles of high-performance Othello agents like Logistello (Buro, 1995), we propose an adaptive Minimax agent capable of adapting to latent winning conditions and diverse board geometries. The agent performs a depth-limited tree search where Minimax state nodes $s$ are evaluated by a dynamic utility function $U(s) = w_{pos}(s) \cdot \text{PosScore}(\text{PosNet}(L), s) + \sum_{i=1}^{3} w_i(s) \cdot f_i(s)$, where $f_i(s)$ represents fundamental sub-utilities including mobility, corner occupancy, and disc difference between the player and the opponent. The positional score (PosScore) is calculated through a weighted piece counter (WPC), which utilizes the positional value table generated by PosNet and the placements of the discs to score the current board state.

Traditional Minimax agents rely on static, hand-crafted utility weights; however, such weights are insufficient when the environmental condition $C$ is latent, necessitating a dynamically trainable utility function. To address this challenge, our agent employs WeightNet, a neural network designed to output the optimal weight vector $\mathbf{w}(s) = [w_{pos}(s), w_1(s), w_2(s), w_3(s)]$ conditioned on the current state. WeightNet takes the current game progress and disc occupancy as inputs, allowing the agent to adjust its strategic priorities dynamically as the game evolves. Similar to PosNet, as the victory condition $C$ is latent and an explicit fitness function cannot be defined, WeightNet is trained via survival-only neuroevolution during the interaction phase. Note that its tree expansion relies only on the fundamental Othello transition rules for generating successor board states, without direct access to the actual environment. Technical details regarding the WeightNet architecture and the evolutionary operators are provided in Appendix D.

### 4.4  Comparative Baselines

To provide a comprehensive evaluation of our adaptive approach, we include several comparative baselines that represent different levels of environmental knowledge and learning efficiency. These baselines serve as reference points for comparing survival-based adaptation with search and learning-based alternatives under different access assumptions.

**Monte Carlo Tree Search (MCTS)** (Coulom, 2006) serves as a *simulator-privileged* reference in our Arena. Unlike other agents that must infer the latent winning condition $C$ from sparse terminal signals through sequential interaction alone, MCTS is granted direct access to the environment simulator during search. By performing simulated playouts governed by the true environment rules—including the latent threshold $K$—MCTS gains statistical access to the hidden objective at decision time without needing to discover it through trial and error. MCTS should therefore not be interpreted as a matched-information baseline, but rather as a reference illustrating the level of performance achievable when the latent objective can be probed through simulation. We evaluate MCTS with 30, 50, and 100 simulations per move; detailed specifications are provided in Appendix E.

**Proximal Policy Optimization (PPO)** (Schulman et al., 2017) is evaluated as a representative of modern gradient-based RL to quantify the efficiency gap relative to evolution-based adaptation. Each PPO agent is trained independently per environment under the same interaction budget as the adaptive Minimax agent. Unlike the adaptive Minimax, which adapts through population-level paired competition, PPO trains a single policy network via self-play against its own copy. We use a masked variant that receives the legal action set $\mathcal{A}_t$ and masks invalid actions during policy optimization. The latent nature of the objectives and the sparsity of terminal-only rewards pose significant challenges for gradient-based convergence within this limited budget. By evaluating PPO under the same budget and interaction protocol, we also examine whether a standard gradient-based method can achieve rapid adaptation without privileged simulator access. Technical specifications regarding the PPO architecture and training setup are detailed in Appendix F.

**Random Agent** selects actions uniformly at random from the valid action set. This baseline serves as a reference for the minimum level of play achievable without any learning or search.

# 5 Results and Analysis

## 5.1 Evaluation of Meta-Learned Spatial Priors

A foundational component of our framework is rooted in generalized prior expertise regarding board topology, as represented by the meta-learned PosNet. Setting aside the complexities of adapting to the latent winning condition $\mathcal{C}$, we first demonstrate that a generalized spatial mapping of strategic importance across diverse board topologies $\mathcal{L}$ is effectively achievable.

Figure 1 illustrates that PosNet is capable of learning generalizable spatial priors using only terminal rewards, showing value tables that logically prioritize stable positions and penalize risky positions. In Layout 1 (classical Othello board), PosNet prioritizes the corners most, which are the non-flippable positions. Furthermore, PosNet penalizes X-squares and the C-Squares, which are the neighbors of corners that risk the opponent getting corner cells when the player's discs are placed on them. Such results align with the general strategic consensus of Othello (Rose, 2005).

Beyond the classical board, PosNet demonstrates meaningful generalization to the other layouts. A representative example of this topological reasoning is observed in Layout 2 (No Corners). Unlike the classical board, where both C-squares and X-Squares are penalized, in Layout 2, the C-squares become non-flippable positions, while X-squares still remain risky, as placing discs on X-squares still allows opponents to take C-squares. PosNet generates the value table that logically follows such changes, as C-squares are prioritized and the X-squares remain penalized.

## 5.2 Adaptation Performance

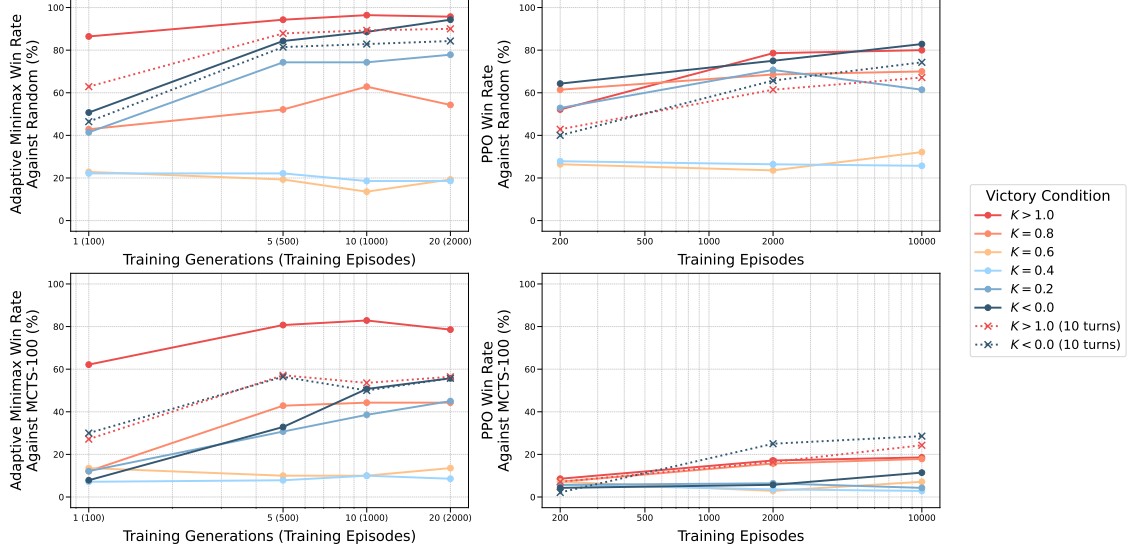

Figure 2: Convergence speed of Adaptive Minimax (left) and PPO (right) under increasing training budgets, measured by evaluation win rate against Random (top) and MCTS-100 (bottom). Each curve is averaged over the 7 environments sharing the same victory condition ($C$). For the Minimax, each generation requires 100 games (see §B.4.). Detailed per-environment results are in §H.

The primary objective of the Expanded Othello AI Arena is to evaluate skill-acquisition efficiency under a limited interaction budget. Figure 2 therefore reports evaluation win rates at different training checkpoints rather than only the final policy after training. Both Adaptive Minimax and PPO are trained through self-play, and the curves show their win rates against external opponents after each amount of training. From this perspective, the 2,000-game budget is informative rather than arbitrary: substantial differences between methods already emerge within this range. In particular, Adaptive Minimax shows rapid early improvement, often reaching a high win rate after only a single generation of updates in the standard, inverse,

| Adaptive Minimax | vs Random | vs PosNet | vs PPO-2K | vs MCTS-30 | vs MCTS-50 | vs MCTS-100 |
|---|---|---|---|---|---|---|
| **Victory Condition** ($C$) | **Win Rate (%)** | | | | | |
| $K > 1.0$ | $95.7 \pm 7.3\%$ | $78.6 \pm 29.8\%$ | $85.7 \pm 37.8\%$ | $85.0 \pm 10.8\%$ | $84.3 \pm 17.2\%$ | $78.6 \pm 18.0\%$ |
| $K = 0.8$ | $54.3 \pm 17.7\%$ | $47.9 \pm 25.0\%$ | $64.3 \pm 24.4\%$ | $47.1 \pm 15.2\%$ | $44.3 \pm 18.1\%$ | $44.3 \pm 25.2\%$ |
| $K = 0.6$ | $19.3 \pm 11.0\%$ | $19.3 \pm 22.4\%$ | $0.0 \pm 0.0\%$ | $14.3 \pm 10.6\%$ | $9.3 \pm 7.3\%$ | $13.6 \pm 14.4\%$ |
| $K = 0.4$ | $18.6 \pm 12.5\%$ | $3.6 \pm 6.3\%$ | $14.3 \pm 24.4\%$ | $7.1 \pm 8.1\%$ | $10.7 \pm 7.9\%$ | $8.6 \pm 12.5\%$ |
| $K = 0.2$ | $77.9 \pm 13.2\%$ | $56.4 \pm 31.8\%$ | $64.3 \pm 47.6\%$ | $60.7 \pm 22.6\%$ | $48.6 \pm 21.0\%$ | $45.0 \pm 19.8\%$ |
| $K < 0.0$ | $94.3 \pm 5.3\%$ | $100.0 \pm 0.0\%$ | $92.9 \pm 18.9\%$ | $65.7 \pm 26.8\%$ | $59.3 \pm 29.2\%$ | $55.7 \pm 24.4\%$ |
| $K > 1.0$ (10 turns) | $90.0 \pm 5.8\%$ | $83.6 \pm 23.2\%$ | $85.7 \pm 24.4\%$ | $65.0 \pm 15.8\%$ | $64.3 \pm 15.1\%$ | $56.4 \pm 25.4\%$ |
| $K < 0.0$ (10 turns) | $84.3 \pm 8.9\%$ | $82.1 \pm 23.8\%$ | $78.6 \pm 26.7\%$ | $61.4 \pm 18.2\%$ | $57.9 \pm 14.7\%$ | $55.7 \pm 17.4\%$ |
| | **Draw Rate (%)** | | | | | |
| $K > 1.0$ | $1.4 \pm 3.8\%$ | $0.7 \pm 1.9\%$ | $0.0 \pm 0.0\%$ | $0.7 \pm 1.9\%$ | $2.1 \pm 3.9\%$ | $0.7 \pm 1.9\%$ |
| $K = 0.8$ | $35.7 \pm 20.9\%$ | $39.3 \pm 35.9\%$ | $28.6 \pm 26.7\%$ | $31.4 \pm 20.6\%$ | $24.3 \pm 15.9\%$ | $20.0 \pm 13.8\%$ |
| $K = 0.6$ | $73.6 \pm 13.8\%$ | $72.1 \pm 22.7\%$ | $92.9 \pm 18.9\%$ | $67.1 \pm 15.5\%$ | $72.9 \pm 11.9\%$ | $67.9 \pm 19.3\%$ |
| $K = 0.4$ | $72.9 \pm 18.0\%$ | $95.7 \pm 7.3\%$ | $78.6 \pm 26.7\%$ | $70.0 \pm 27.2\%$ | $54.3 \pm 30.1\%$ | $60.0 \pm 27.1\%$ |
| $K = 0.2$ | $16.4 \pm 13.5\%$ | $43.6 \pm 31.8\%$ | $14.3 \pm 37.8\%$ | $16.4 \pm 17.7\%$ | $17.9 \pm 16.5\%$ | $24.3 \pm 15.9\%$ |
| $K < 0.0$ | $2.1 \pm 3.9\%$ | $0.0 \pm 0.0\%$ | $0.0 \pm 0.0\%$ | $1.4 \pm 2.4\%$ | $5.7 \pm 9.3\%$ | $3.6 \pm 4.8\%$ |
| $K > 1.0$ (10 turns) | $6.4 \pm 7.5\%$ | $9.3 \pm 18.4\%$ | $14.3 \pm 24.4\%$ | $24.3 \pm 15.1\%$ | $17.9 \pm 11.5\%$ | $32.1 \pm 28.4\%$ |
| $K < 0.0$ (10 turns) | $12.1 \pm 7.6\%$ | $15.0 \pm 24.0\%$ | $14.3 \pm 24.4\%$ | $20.0 \pm 14.4\%$ | $17.1 \pm 13.2\%$ | $10.7 \pm 9.8\%$ |
| | **Lose Rate (%)** | | | | | |
| $K > 1.0$ | $2.9 \pm 3.9\%$ | $20.7 \pm 29.9\%$ | $14.3 \pm 37.8\%$ | $14.3 \pm 9.8\%$ | $13.6 \pm 14.1\%$ | $20.7 \pm 17.2\%$ |
| $K = 0.8$ | $10.0 \pm 14.4\%$ | $12.9 \pm 19.1\%$ | $7.1 \pm 18.9\%$ | $21.4 \pm 22.7\%$ | $31.4 \pm 29.5\%$ | $35.7 \pm 28.3\%$ |
| $K = 0.6$ | $7.1 \pm 9.5\%$ | $8.6 \pm 18.6\%$ | $7.1 \pm 18.9\%$ | $18.6 \pm 8.0\%$ | $17.9 \pm 9.5\%$ | $18.6 \pm 15.7\%$ |
| $K = 0.4$ | $8.6 \pm 8.5\%$ | $0.7 \pm 1.9\%$ | $7.1 \pm 18.9\%$ | $22.9 \pm 27.5\%$ | $35.0 \pm 31.2\%$ | $31.4 \pm 31.1\%$ |
| $K = 0.2$ | $5.7 \pm 6.7\%$ | $0.0 \pm 0.0\%$ | $21.4 \pm 39.3\%$ | $22.9 \pm 22.0\%$ | $33.6 \pm 24.1\%$ | $30.7 \pm 22.3\%$ |
| $K < 0.0$ | $3.6 \pm 3.8\%$ | $0.0 \pm 0.0\%$ | $7.1 \pm 18.9\%$ | $32.9 \pm 27.1\%$ | $35.0 \pm 30.1\%$ | $40.7 \pm 25.1\%$ |
| $K > 1.0$ (10 turns) | $3.6 \pm 2.4\%$ | $7.1 \pm 18.9\%$ | $0.0 \pm 0.0\%$ | $10.7 \pm 6.1\%$ | $17.9 \pm 9.9\%$ | $11.4 \pm 6.3\%$ |
| $K < 0.0$ (10 turns) | $3.6 \pm 4.8\%$ | $2.9 \pm 7.6\%$ | $7.1 \pm 18.9\%$ | $18.6 \pm 12.8\%$ | $25.0 \pm 14.1\%$ | $33.6 \pm 15.7\%$ |

Table 1: Average win, draw, and lose rates (%) of the adaptive Minimax for each victory condition ($C$), reported as mean with standard deviation. Each entry is averaged over the 7 environments that share the same $C$, i.e., over 7 different layouts ($L$). The adaptive Minimax is trained for 20 generations with a population size of 50. See §B.4 for budget details, and Table 2 for environment-specific win, draw, and lose rates.

and time-limited regimes. This rapid convergence stems from its population-level survival mechanism, which extracts an actionable signal from sparse terminal outcomes without requiring gradient information. By contrast, PPO lacks sufficient gradient signal under sparse terminal rewards, and its win rate curves plateau early and remain largely flat even at 10,000 games (Figure 2, right), and the Adaptive Minimax wins 85–93% of head-to-head matches against PPO-2K in the standard and inverse regimes (Table 1).

Table 1 complements this convergence view by summarizing the final Win/Draw/Lose profile of Adaptive Minimax after 20 generations (2,000 games) of training. The results show that the difficulty of adaptation is strongly regime-dependent. In the standard, inverse, and 10-turn conditions, Adaptive Minimax generally converts adaptation into high win rates against most opponents. Notably, in broad regimes, the Adaptive Minimax achieves win rates comparable to MCTS-100 (e.g., 78.6% for $K > 1.0$), despite MCTS having privileged simulator access to the latent threshold $K$ during search. In contrast, the narrow-interval regimes remain much harder, but not uniformly so: the most difficult cases are $K = 0.6$ and $K = 0.4$, whereas $K = 0.8$ and $K = 0.2$ remain substantially more manageable. This suggests that the main challenge is not merely the presence of a hidden interval, but the need to control the terminal occupancy ratio with high precision when the admissible range becomes narrow and lies close to the majority/minority boundary.

The draw statistics are particularly important for interpreting these narrow-regime results. Under the outcome structure, low win rates do not necessarily indicate immediate collapse into losses. Instead, for $K = 0.6$ and $K = 0.4$, the final outcomes are often dominated by draws, indicating that the agent frequently learns to avoid clearly losing terminal occupancies even when it fails to enter the winning interval reliably. In this sense, adaptation in narrow regimes often appears first as conservative terminal-ratio control rather than consistent win conversion. Taken together, the results suggest that the Arena captures two related but distinct aspects

of adaptation under limited interaction: rapid win-oriented convergence in broad regimes, and more fragile, draw-heavy control behavior in narrow regimes where precise terminal regulation is required.

# 6 Future Directions for Research with the Expanded Othello AI Arena

The empirical analysis in Section 5 demonstrated that while the adaptive Minimax agent achieves competitive performance in several environments, it continues to struggle in narrow victory regimes that require precise terminal occupancy control under latent objectives. Building upon these findings, this section outlines future research directions that can be facilitated by the Expanded Othello AI Arena.

**Rapid Adaptation via Prior Knowledge and Test-Time Learning.** A key open question is how broad prior knowledge—whether encoded in foundation models (Wei et al., 2022; Achiam et al., 2023) or acquired through meta-learning—can be made actionable for discovering latent objectives from sparse terminal signals. Current frontier models struggle with online adaptation to unseen interactive tasks, scoring below 1% on benchmarks that require it (ARC Prize Foundation, 2026). The Arena offers a controlled setting to study this gap, where test-time parameter optimization (Sun et al., 2020; 2024) or in-context few-shot adaptation (Raparthy et al., 2024; Feng et al., 2025) could treat game trajectories and terminal rewards as demonstration samples for rapid strategy synthesis.

**Factoring Dynamics from Objectives in World Models.** Recent implicit world models learn latent representations that predict outcomes rather than reconstructing observations, achieving high sample efficiency across diverse control tasks (Schrittwieser et al., 2020; Hansen et al., 2024). A key open question is how to disentangle invariant transition dynamics from task-specific objectives within such models. Reward-free pretraining approaches learn dynamics models that can be adapted to new objectives by modifying only the reward or value components (Sekar et al., 2020), while multitask world models condition all components on learnable task embeddings to implicitly capture inter-task structure (Hansen et al., 2024). The Arena's explicit separation of fixed board dynamics from latent victory conditions $C$ offers a controlled testbed at this intersection: a world model must internalize the invariant Othello transition mechanics shared across all 56 environments while rapidly adapting its objective predictions to the hidden threshold $K$ within the 2,000-game budget—a factorization challenge that neither purely reward-free nor purely embedding-based approaches have been specifically designed to address.

**Adaptive Strategy Revision via Opponent Modeling.** The Arena's zero-sum structure naturally raises the question of whether explicit opponent modeling can improve adaptation, particularly in narrow regimes where the opponent's actions directly affect whether the agent's terminal occupancy falls within the admissible interval. Recent advances in Theory of Mind for AI agents (Sap et al., 2022; Bard et al., 2020) suggest a promising path: developing agents that continuously revise their strategies by modeling and anticipating the opponent's evolving play, rather than treating the adversary as a static component of the environment. The Arena offers a controlled setting for this direction, where future variants could introduce more stable adaptive opponents or richer strategic signaling to make such recursive reasoning increasingly central to successful play.

# 7 Related Works

## 7.1 Benchmarks for Artificial General Intelligence

A prominent recent direction in AGI-oriented evaluation is to emphasize few-shot skill acquisition rather than broad memorized knowledge. The ARC benchmark series (Chollet, 2019; Chollet et al., 2024; 2025) exemplifies this philosophy through progressively challenging static grid-based reasoning tasks. Most recently, ARC-AGI-3 (ARC Prize Foundation, 2026) makes a qualitative shift from static input–output puzzles to interactive, multi-stage game environments where agents must explore, infer goals, and adapt without explicit instructions. Each game introduces novel mechanics at successive stages, and games differ fundamentally in genre and structure—grounded only in shared core knowledge priors such as objectness and basic geometry. Frontier models currently score below 1%, revealing that vast prior knowledge alone does not yield efficient online adaptation to truly unseen interactive tasks. Our Arena shares ARC-AGI-3's core emphasis on online adaptation under latent objectives, efficiency-based evaluation, and the requirement to infer win conditions

from interaction alone. However, the two benchmarks probe complementary facets of the adaptation challenge. ARC-AGI-3 evaluates breadth of adaptation across a diverse collection of single-player games with widely varying mechanics, where the primary difficulty lies in encountering a fundamentally new game at each evaluation instance. The Arena instead isolates depth of adaptation within a single game substrate under adversarial play: the transition dynamics remain fixed, but the latent winning condition $C$ varies across environments, requiring agents to precisely control terminal outcomes against an opponent who co-determines the result. This zero-sum competitive dimension—absent from single-player benchmarks—introduces a distinct challenge where successful adaptation demands not only rule discovery but also strategic adjustment relative to the opponent's behavior.

In parallel, many general capability benchmarks target complementary axes such as breadth of academic knowledge (Hendrycks et al., 2021), broad task coverage (Srivastava et al., 2023), holistic multi-metric evaluation (Liang et al., 2022), or expert-level science QA (Rein et al., 2024). On the environment side, XLand-MiniGrid (Nikulin et al., 2024) emphasizes adaptation across procedurally varied tasks in a scalable meta-RL setting, though without competitive play or latent terminal objectives.

### 7.2 Adaptive Game Playing under Latent Rules and Outcome-Only Feedback

A substantial body of work studies agents that can operate across many games rather than being tailored to a single fixed specification. In General Game Playing (GGP), agents receive an explicit formal game description and must quickly synthesize effective play, motivating standardized languages, infrastructures, and competitions (Genesereth et al., 2005). Similarly, the General Video Game AI (GVGAI) framework evaluates agents across unseen games and emphasizes general-purpose decision-making and planning under limited game-specific knowledge (Perez-Liebana et al., 2019). General game systems such as Ludii further broaden the scope of cross-game evaluation by providing efficient, human-readable game descriptions and a large library of games (Piette et al., 2020). Related interactive environments, such as XLand-MiniGrid, also study adaptation under large-scale procedural variation, although they do not focus on competitive play with latent terminal objectives (Nikulin et al., 2024).

Complementary to general playing given rules, another line of research considers the rule discovery: inferring governing mechanics or objectives from observations of gameplays (Björnsson, 2012). On the algorithmic side, classical game AI has long combined search with learned or tuned evaluation components (Samuel, 1959), and even in board games such as Othello, prior work explored evolving neural networks to focus Minimax search (Moriarty & Miikkulainen, 1994); more broadly, competitive coevolution provides a natural framing for adversarial settings where improvement is measured relative to opponents rather than dense intermediate supervision (Klijn & Eiben, 2021). Building on these perspectives, the Expanded Othello Framework instantiates a controlled family of adversarial environments where the board layout is observed, but the win condition is latent, enabling direct measurement of how efficiently an agent can identify the objective and acquire effective strategies under a strict interaction budget.

## 8 Conclusions

The Expanded Othello AI Arena focuses on the rapid strategic convergence of agents under latent objectives by imposing a strict 2,000-game interaction budget. By decoupling the observable layout ($L$) from the latent winning conditions ($C$), the framework formalizes and parameterizes various expansions of Othello within a unified structure. Furthermore, it compels agents to discern valid victory paths through sequential interaction based solely on sparse terminal feedback. Our empirical analysis shows that this setting distinguishes broad regimes, in which agents can often achieve rapid win-oriented adaptation, from narrow regimes, in which the main challenge shifts toward precise control of the terminal occupancy ratio under hidden objectives. In these narrow settings, adaptation may manifest not only as successful win conversion but also as draw-heavy conservative behavior that avoids clearly losing terminal states.

To further enhance the benchmark's discriminative power, the environment can be expanded to perturb the fundamental mechanisms currently held as constants. While the Arena successfully varies layouts and winning thresholds, the core flipping logic and alternating turn structure remain fixed parameters. Future iterations

can introduce consecutive turns or modified obstacle rules, such as allowing discs to be flipped through immutable barriers, to test whether agents can adapt not only to latent objectives but also to changes in the underlying transition dynamics. Such expansions can be utilized for verifying whether a system can maintain logical decision-making under arbitrary transition dynamics without relying on a fixed game substrate.

## Acknowledgements

This work was supported by NRF (RS-2024-00451162, 60%), IITP (RS-2023-00216011; 20%, AI Graduate School; No. 2019-0-01842; 20%), Ministry of Science and ICT, Korea. Computing resource were partially supported by KISTI.

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

# A Strategies and Heuristics for Classical Othello

## A.1 Human Strategies for Othello

| | C | | | | | C | |
|---|---|---|---|---|---|---|---|
| C | X | | | | | X | C |
| | | | | | | | |
| | | | W | B | | | |
| | | | B | W | | | |
| | | | | | | | |
| C | X | | | | | X | C |
| | C | | | | | C | |

| 1.00 | -0.25 | 0.10 | 0.05 | 0.05 | 0.10 | -0.25 | 1.00 |
|---|---|---|---|---|---|---|---|
| -0.25 | -0.25 | 0.01 | 0.01 | 0.01 | 0.01 | -0.25 | -0.25 |
| 0.10 | 0.01 | 0.05 | 0.02 | 0.02 | 0.05 | 0.01 | 0.10 |
| 0.05 | 0.01 | 0.02 | 0.01 | 0.01 | 0.02 | 0.01 | 0.05 |
| 0.05 | 0.01 | 0.02 | 0.01 | 0.01 | 0.02 | 0.01 | 0.05 |
| 0.10 | 0.01 | 0.05 | 0.02 | 0.02 | 0.05 | 0.01 | 0.10 |
| -0.25 | -0.25 | 0.01 | 0.01 | 0.01 | 0.01 | -0.25 | -0.25 |
| 1.00 | -0.25 | 0.10 | 0.05 | 0.05 | 0.10 | -0.25 | 1.00 |

Figure 3: (Left) Locations of the C-squares (C), X-squares (X), and the initial black (B) and white (W) disc positions. (Right) The positional value table of the standard weighted piece counter heuristic agent created by Yoshioka et al. (1999).

There exists a set of generally accepted consensus strategies among the Othello experts, which is used as the fundamental guides for the classical Othello game (Rose, 2005). A central concept is the *stability*, finding the non-flippable cells. Most notably, discs placed on corners are not flippable for the rest of the game, and corner control often anchors stable edges that are difficult to challenge. Accordingly, strong play prioritizes securing corners when possible, while carefully avoiding moves that allow opponents to gain the corners.

Two well-known risky regions are the *C-squares* and *X-squares*, which illustrated in Figure 3 (Left). The C-square is the orthogonally adjacent cell to a corner, and the X-square is the diagonally adjacent cell. Placing a disc on these cells is often dangerous because it can enable the opponent to immediately capture the corner. As a result, C- and X-squares are typically penalized unless the corresponding corner is already secured or a concrete tactical justification exists.

Another major principle is *mobility* — the number of legal moves available. Maintaining high mobility while restricting the opponent's mobility improves positional control and reduces the risk of being forced into unfavorable moves, especially those that open access to corners and stable edges. On the other hand, early-game disc count is not a reliable objective as Othello shows high board state changes, due to its signature flipping mechanism. Hence, strong players often accept temporary material loss to preserve mobility and long-term stability, while disc advantage becomes decisive mainly in the late game.

## A.2 Heuristic Agents

Before modern learning-based agents, strong Othello play was commonly built on hand-designed (or statistically fitted) evaluation functions coupled with adversarial search. A canonical family is the *positional heuristic* that assigns a fixed weight to each board square and evaluates a position by a weighted sum over occupied squares, often called a *weighted piece counter* (WPC), which calculates a value of the state $s$ given the positional value table $V$, following Equation 5. Note that $s_{i,j} = 1$ if player disc is on the cell $(i, j)$, $-1$ if opponent's disc is on the cell, and 0 if empty.

$$\text{WPC}(V, s) = \sum_{i=1}^{W} \sum_{j=1}^{H} V_{i,j} s_{i,j} \tag{5}$$

Classic positional value tables typically emphasize stable regions such as corners and penalize risky near-corner squares (e.g., C- and X-squares), encoding well-known human principles in a simple linear form (Yoshioka et al., 1999). Despite its simplicity, WPC-style evaluation has served as a long-standing baseline and a convenient expert proxy in Othello research, especially when paired with shallow Minimax search.

A major milestone in heuristic Othello was *Logistello*, which demonstrated that high-level performance can be achieved by combining deep search with evaluation functions learned from data. Logistello popularized constructing evaluation functions from large collections of game positions and outcomes, fitting parameters statistically rather than solely relying on hand-tuned tables (Buro, 1995). In practice, Logistello-style approaches can be viewed as a progression from fixed positional heuristics (static WPC tables) to richer feature sets and statistically optimized weights, with search (e.g., Minimax/alpha–beta) acting as the decision-time optimizer. This line of work remains directly relevant to our setting because it clarifies what is achievable with strong *positional* priors and search-based planning, even before introducing more adaptive mechanisms.

# B Detailed Procedures of Survival-Only Evolutionary Learning

This appendix details the survival-only evolutionary procedure used throughout our framework. The key constraint is that the winning condition $C$ is latent, so we cannot define a reliable scalar fitness or shaped reward. Hence, the only trustworthy signal is the terminal match outcome. Accordingly, we adopt the *survival-only* selection mechanism in which individuals compete against each other and only the winners are duplicated, followed by mutation and reshuffling. This procedure is used (i) to meta-learn PosNet parameters over layout distributions, and (ii) to train environment-specific WeightNet parameters of the adaptive Minimax within the 2,000-game interaction budget. Algorithm 1 summarizes the survival-only evolution, which is explained in detail from Section B.1 to Section B.4.

---

**Algorithm 1** Survival-only evolution

---

**Require:** Population size $2N$, generations $G$, mutation scale $\sigma$, environment $E$
 1: Initialize population $P = \{\theta_i\}_{i=1}^{2N}$
 2: **for** $g = 1$ to $G$ **do**
 3:     Instantiate agents $\{A_i\}_{i=1}^{2N}$ from $\{\theta_i\}$ in environment $E$
 4:     Pair individuals $(i, i + N)$ for $i = 1, \ldots, N$
 5:     For each pair, play two games with swapped colors and compute $\{\text{match}(A_i, A_{i+N})\}_{i=1}^{N}$
 6:     Construct selected multiset $S$ by Eq. (6)
 7:     Mutate: $P_{\text{mut}} = \{\theta'_i + \epsilon_i\}$ with $\epsilon_i \sim \mathcal{N}(0, \sigma^2 I)$
 8:     Shuffle: $P \leftarrow \text{Shuffle}(P_{\text{mut}})$
 9: **end for**
10: **return** final population $P$ (to be ensembled/normalized as in Appendix B.3)

---

### B.1 Survival-Only Selection via Paired Matches

Let $P = \{\theta_i\}_{i=1}^{2N}$ denote a population of size $2N$ (even-sized for pairing). Each $\theta_i$ parameterizes an instantiated agent (e.g., neural network parameter weights of PosNet and WeightNet). We evaluate individuals via paired matches. Let $A_i$ denote an agent with $\theta_i$. Then, for each pair $(i, i+N)$, we play two games between $A_i$ and $A_{i+N}$, with swapped colors to reduce first-move bias. Let $\text{match}(A_i, A_j) \in \{1, 0, -1\}$ be the outcome from $A_i$'s perspective (victory/draw/defeat). We then construct a *selected* multiset $S$ of size $2N$ by duplicating the winner:

$$S = \bigcup_{i=1}^{N} \begin{cases} \{\theta_i, \theta_i\} & \text{if } \text{match}(A_i, A_{i+N}) = 1, \\ \{\theta_{i+N}, \theta_{i+N}\} & \text{if } \text{match}(A_i, A_{i+N}) = -1, \\ \{\theta_i, \theta_{i+N}\} & \text{if } \text{match}(A_i, A_{i+N}) = 0. \end{cases} \tag{6}$$

Intuitively, this is a pure-survival scheme; selection pressure is induced only by head-to-head outcomes, without assuming any explicit fitness function.

### B.2 Mutation, Shuffle, and Generational Update

After selection, each retained parameter vector is mutated by additive Gaussian noise:

$$P_{\text{mut}} = \{\theta_i' + \epsilon_i\}_{i=1}^{2N}, \qquad \epsilon_i \sim \mathcal{N}(0, \sigma^2 I). \tag{7}$$

We then randomly shuffle the mutated population to form the next generation

$$P \leftarrow \text{Shuffle}(P_{\text{mut}}), \tag{8}$$

and repeat the cycle (agent instantiation $\rightarrow$ paired matches $\rightarrow$ comparison-only selection $\rightarrow$ mutation $\rightarrow$ shuffle) for $G$ generations.

### B.3 Ensembling and Normalization at Inference

We use a population ensemble at the end of evolution to reduce variance within the agents in the population $P$. The concrete aggregation depends on which object is being evolved:

**PosNet / positional value tables.** Each individual $\theta_i$ induces a value table $V_i$ over the board. We normalize each table by affine rescaling so that its minimum becomes $-1$ and maximum becomes $1$, then average the normalized tables to obtain the final table $\bar{V}$ used by the positional baseline (PosNet).

**WeightNet / utility re-weighting.** Each individual $\theta_i$ induces a WeightNet that outputs a weight vector $w_i(s)$ for a given state $s$. For stability, we normalize each vector by an $L_1$-style normalization, $\text{Normalize}(w) = w / \sum_k |w_k|$, and use the mean normalized weights $\bar{w}(s) = \frac{1}{2N} \sum_{i=1}^{2N} \text{Normalize}(w_i(s))$ over the final population.

### B.4 Budget accounting and default hyperparameters

All training in a single environment $E$ is constrained by a fixed interaction budget ($N_{games} = 2000$). For the adaptive Minimax, this corresponds to $G = 20$ generations with population size $N = 50$ (thus $2N = 100$ individuals), where each selection pairing uses two games (side-swapped) per match, resulting in 100 games per generation, yielding 2,000 total games across 20 generations.

## C Detailed Procedures of PosNet

This section explains the detailed procedure of crafting the positional network (PosNet) to meta-learn spatial prior knowledge from diverse layouts ($L$). Figure 4 shows an example procedure of the value table ($V$) generation using PosNet.

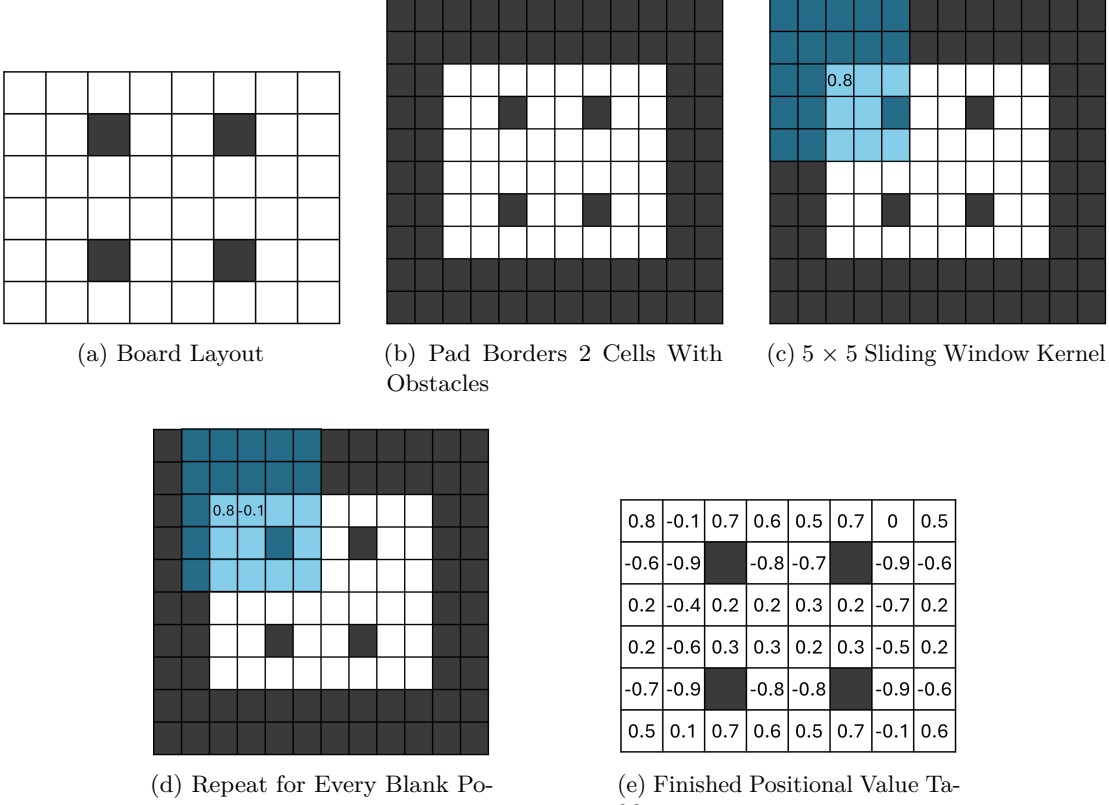

(a) Board Layout

(b) Pad Borders 2 Cells With Obstacles

(c) 5 × 5 Sliding Window Kernel

(d) Repeat for Every Blank Position

(e) Finished Positional Value Table

Figure 4: An example procedure of the positional value table generation.

### C.1 Network Architecture and Input Representation

The positional network (PosNet) is implemented as a lightweight fully-convolutional network (FCN), specifically engineered to generate positional value tables $V$ for any board layout $L \in \mathcal{L}$. Unlike traditional Othello agents that rely on fixed $8 \times 8$ tables, the FCN structure allows PosNet to process boards of arbitrary geometries — such as those found in our random configuration layouts (Layouts 5–7) — by applying the same learned spatial filters across the entire grid.

The architecture comprises a convolutional layer with $5 \times 5$ kernels, followed by a final $1 \times 1$ convolutional layer. We purposefully selected a $5 \times 5$ kernel size based on the following strategic and technical rationales:

- **Local Context for Heuristic Identification**: In classical Othello, the strategic value of a square is primarily defined by its proximity to corners and edges. The most critical squares (C-squares and X-squares) are located exactly 1 cell (orthogonally or diagonally) away from a corner. Furthermore, as the expert-generated value table in Figure 3 (Right) shows, the major cell value differences occur within the 2 cell distances from the corners or the edges. The $5 \times 5$ window provides a 2-cell radius, which is the minimum sufficient area to perceive a corner or edge from any of these critical positions.

- **Inductive Bias of Spatial Stability**: Human expertise (Rose, 2005) suggests that stability — the inability of a disc to be flipped — is the most important factor in positional valuing. Since stability is a property determined by the local topology (e.g., being in a corner), the $5 \times 5$ kernel acts as an inductive bias that forces the network to learn these local patterns rather than memorizing global board states.

- **Architectural Compatibility for Variable Geometries**: By utilizing a fully-convolutional architecture devoid of fixed-size components such as fully-connected layers, PosNet remains inherently

invariant to the input dimensions. This design ensures that the meta-learned spatial filters can be applied zero-shot to any board size ($W \times H$) or the placement of the obstacles.

To ensure consistent behavior at the board boundaries, we apply a 2-cell padding of obstacles around the input. This ensures that the $5 \times 5$ kernel always has full context, even when evaluating squares on the very edge of the board, effectively treating out-of-bound cells as impassable obstacles. Note that we chose a $5 \times 5$ kernel rather than $4 \times 4$ since an odd-sized kernel provides a unique center cell for a square-centered symmetric receptive field.

## C.2 Stabilization via Orthogonal Transformation

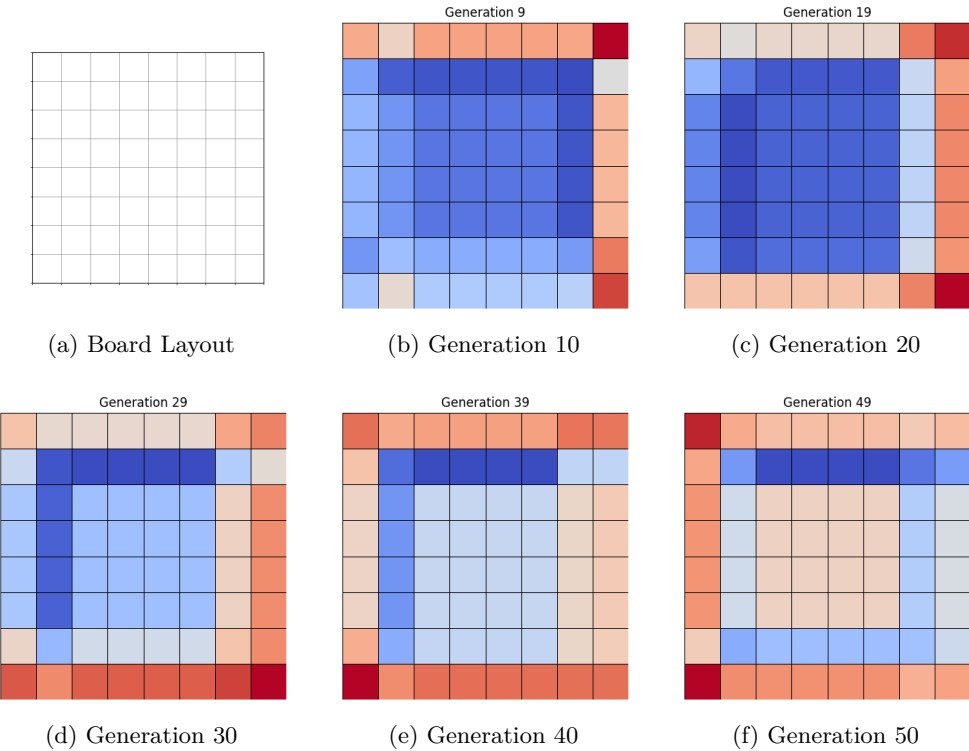

Figure 5: Positional Value Table from PosNet trained separately on the standard Othello ($8 \times 8$) environment ($K > 1.0$), without meta-learning or stabilization process. A population size of 50 is used, and the agents are trained for 50 generations. Red means higher positional values, while blue means lower values. The drifting effect is evidently shown; directional imbalance can be observed.

To counteract the *drifting* effect (e.g., Figure 5) — where comparison-only selection leads to unstable directional biases — we employ a rigorous stabilization protocol.

- **Symmetric Nature of Othello**: Conceptually, an Othello board has no inherent orientation. Hence, PosNet's output should ideally be equivariant to orthogonal transformations. Consider an orthogonal transformation $F$ from the eight-fold symmetry group (D4 dihedral group), which consists of rotations and reflections. Due to orthogonal transformation equivalence, Equation 9 should hold.

$$F(\text{PosNet}(L)) = \text{PosNet}(F(L)) \tag{9}$$

Then, for some orthogonal transformation $F$ and its inverse function $F^{-1}$, Equation 10 should hold.

$$\text{PosNet}(L) = F^{-1}(\text{PosNet}(F(L))) \tag{10}$$

- **Training and Inference Protocol**: During meta-learning, each agent in the population is evaluated under a randomly chosen transformation $F$, forcing the weights to generalize across all orientations. Hence, the positional value table $V$ is generated as Equation 11.

$$V_{\text{train}} = F^{-1}(\text{PosNet}(F(L))) \tag{11}$$

At inference time, we apply all eight transformations to the layout $L$, pass them through PosNet, and average the inverse-transformed outputs to produce a symmetrical value table, following Equation 12.

$$V_{\text{infer}} = \frac{1}{8} \sum_{i=1}^{8} F_i^{-1}(\text{PosNet}(F_i(L))) \tag{12}$$

## C.3 Meta-Learning and Spatial Generalization

PosNet evolves via a meta-learning scheme, using the survival-only evolution over a distribution of 1,000 randomly generated board layouts. By keeping the winning condition fixed ($K > 1.0$) during this phase to isolate the spatial reasoning from the influence of variant winning conditions ($C$), the network learns a universal spatial prior — interpretation of stable cells and risky cells on an arbitrary layout $L$ — which then serves as both a robust foundation for the adaptive Minimax agent during environment-specific adaptation and the non-adapting positional baseline in the experiments. Note that the PosNet is trained for 1,000 generations with a population size of 200.

# D Detailed Procedures of Adaptive Minimax

## D.1 Adaptive Utility Function and Sub-Utilities

Motivated by Logistello (Buro, 1995), the proposed adaptive agent baseline utilizes a Minimax search algorithm with alpha-beta pruning, governed by a dynamic utility function $U(s)$. To enable adaptation to unknown winning conditions $C$, the utility is defined as a learnable linear combination of four fundamental sub-utility functions following Equation 13.

$$U(s) = w_{\text{pos}} \cdot \text{PosScore}(\text{PosNet}(L), s) + w_{\text{mobility}} \cdot \text{Mobility}(s) + w_{\text{corner}} \cdot \text{Corner}(s) + w_{\text{disc}} \cdot \text{discDiff}(s) \tag{13}$$

The weights $\mathbf{w} = [w_{\text{pos}}, w_{\text{mobility}}, w_{\text{corner}}, w_{\text{disc}}]$ are dynamically generated by WeightNet based on the game state. The sub-utilities $u$ are defined as follows:

- **PosScore**: The positional value score of state $S$, calculated through WPC, using the value table generated by the meta-learned PosNet, following Equation 14. Note that $s_{i,j} = 0$ for obstacles.

$$\text{PosScore}(\text{PosNet}(L), s) = \frac{1}{n_{agent} + n_{opponent}} \text{WPC}(\text{PosNet}(L), s) \tag{14}$$

- **Mobility**: The difference between the number of legal moves available to the agent and the opponent. Let $\phi_i(s)$ denote the number of valid moves for player $i$ in state $s$. Then the mobility is defined as Equation 15.

$$\text{Mobility}(s) = \frac{\phi_{agent}(s) - \phi_{opponent}(s)}{\phi_{agent}(s) + \phi_{opponent}(s)} \tag{15}$$

- **Corner**: The difference between the number of corners occupied by the agent and the opponent, following Equation 16.

$$\text{Corner}(s) = \frac{\#\text{Corners Occupied by Agent} - \#\text{Corners Occupied by Opponent}}{4} \tag{16}$$

- **discDiff**: The difference between the number of the agent's discs and the opponent's discs, following Equation 17.

$$\text{discDiff}(s) = \frac{n_{agent} - n_{opponent}}{n_{agent} + n_{opponent}} \tag{17}$$

To maintain numerical stability across diverse board configurations, all sub-utilities are normalized to a range of $[-1, 1]$.

## D.2 WeightNet and State Representation for Strategy Adaptation

WeightNet is a shallow MLP designed for rapid strategy synthesis within the restricted 2,000-game interaction budget. Its objective is to map the temporal and zero-sum competitive context to gain an optimal weight vector $\mathbf{w}$, following Equation 18.

$$\mathbf{w} = \text{WeightNet}(\text{progress}, \rho, 1 - \rho) \tag{18}$$

The input features provide a functional basis for environmental adaptation:

- **Progress**: The ratio of total discs to the board area, representing the progress of the game, defined as Equation 19.

$$\text{progress} = \frac{n_{agent} + n_{opponent}}{W \times H} \tag{19}$$

- **Occupancy ratios** ($\rho$, $1 - \rho$): Following Equation 3, $\rho$ represents the agent's disc occupancy and $1 - \rho$ represents the opponent's disc occupancy.

Such state representation works as a compact basis for environment-specific adaptation under latent objectives.

## D.3 Online Adaptation via Neuroevolution

WeightNet is trained independently for each environment $E$ using the survival-only neuroevolutionary framework. For each environment $E$, the WeightNet is trained with a population size of 100, for 20 generations, i.e., $2000 = \#\text{generations} \times \#\text{games per generation} = 20 \times (100 \times 0.5 \times 2)$. During the training phase, the agent instantiation is done via generating an adaptive Minimax agent for each WeightNet in the population.

The adaptive Minimax uses a depth-3 Minimax search during training for computational efficiency. For the evaluation, the depth is increased to depth-5 to rigorously assess the converged policy $\pi_E$ with the ensemble formed by averaging the normalized weights of the population from the final generation.

Regarding the search procedure of Minimax, the adaptive Minimax uses a deterministic transition model for observable board state transition only; it can search over legal moves and resulting board states under the fixed Othello flipping mechanics (§2.2), but unlike MCTS, it has no access to the actual Expanded Othello environment simulator.

# E   Specifications of MCTS agents

## E.1 Simulation-Based Look-Ahead Baseline

In the Expanded Othello AI Arena, Monte Carlo Tree Search (MCTS) serves as a computation-intensive, rollout-based reference method. Unlike other adaptive agents in this benchmark, MCTS is evaluated with privileged access to the environment simulator during search. It does not receive the latent threshold $K$ as an explicit input; instead, it probes the latent objective indirectly by performing simulated playouts under the true environment dynamics and terminal outcome conditions. Through these rollouts, MCTS obtains a statistical look-ahead into terminal rewards that already reflect the hidden winning rule. Accordingly, MCTS should be interpreted as a simulator-privileged reference rather than a matched-information baseline.

## E.2 Tree Search Mechanism

The MCTS agents follow the standard four-stage cycle to navigate the game tree:

1. **Selection**: Starting from the root, the agent traverses the tree by selecting child nodes that maximize the upper confidence bound for trees (UCT) (Kocsis & Szepesvári, 2006), following Equation 20.

$$\text{UCT}(s, a) = \bar{Q}(s, a) + c\sqrt{\frac{\ln N(s)}{n(s, a)}} \tag{20}$$

   where $\bar{Q}(s, a)$ is the average reward (i.e., average win rate in case of the Expanded Othello) from simulations, $N(s)$ is the total visit count of the parent, $n(s, a)$ is the visit count of the child, and $c$ is the exploration constant.

2. **Expansion**: If a selected node is not terminal and has unvisited legal moves, a new child node is added to the tree.

3. **Rollout**: From the expanded node, the agent performs a random playout to evaluate potential trajectories. These simulations are executed using the environment's ground-truth rules and $K$, enabling the agent to sample potential rewards from the resulting state transitions during the search process.

4. **Backpropagation**: The final outcome is propagated back up the tree to update the aveage win rates ($\bar{Q}(s, a)$) and node visit counts of all ancestral nodes.

### E.3 Implementation Details

To provide diverse reference points for computational budgets, we evaluate three variants: MCTS-30, MCTS-50, and MCTS-100, where the number denotes the total simulations performed per move. For the exploration constant, we set $c = 1.4 \approx \sqrt{2}$. For the rollout depth limit, each simulation is limited to a maximum depth of 30 steps to ensure computational efficiency. If the game does not terminate within this limit, the state is evaluated based on the current occupancy ratio $\rho$ relative to $K$. At the conclusion of the search, the agent selects the action $a$ that maximizes the visit count $N(s, a)$ rather than the average win rate, ensuring robustness against stochastic simulation noise and outliers.

## F Specifications of PPO agents

The Proximal Policy Optimization (PPO) agents are implemented using the `MaskablePPO` implementation built on `stable-baselines3` (Raffin et al., 2021), with the default `MlpPolicy` configuration. A unique PPO agent is initialized and trained independently for each environment $E$ in the Arena to ensure that no cross-environmental transfer or prior knowledge influences the learning process. Each agent is trained for the same 2,000-game interaction budget as the adaptive Minimax agent, using self-play. The agent receives the legal action set $\mathcal{A}_t$, and invalid actions are masked during policy optimization. The reward structure is strictly terminal, with a reward of 1.0 for a victory, -1.0 for a defeat, and 0.0 for a draw, provided only at the termination of a game (§2.4). This configuration serves as a baseline to measure the performance of a standard gradient-based RL method under the same sequential interaction budget, without privileged simulator access or environment-specific prior knowledge.

## G Further Analysis of the Results

In this section, we further examine whether the adaptive Minimax develops occupancy-sensitive adaptation in narrow-regime environments by analyzing how the output $\mathbf{w}$ of WeightNet changes with respect to the player disc occupancy $\rho$. Recall that WeightNet receives a three-dimensional context vector as input: the game progress (Eq. 19), the agent's disc occupancy ratio $\rho = n_{\text{agent}}/(n_{\text{agent}} + n_{\text{opponent}})$, and the opponent's occupancy $1 - \rho$. To probe the learned behavior, we fix the progress to a specific value (0.2, 0.4, 0.6, or 0.8) and sweep $\rho$ from 0 to 1, recording the four output weights $\mathbf{w} = [w_{\text{pos}}, w_{\text{mobility}}, w_{\text{corner}}, w_{\text{disc}}]$. Each subplot in Figures 6 and 7 therefore shows how the utility weights change as a function of the agent's occupancy ratio at a fixed stage of the game. Rather than treating narrow-regime performance simply as a success-or-failure

outcome, our goal here is to investigate whether the learned utility reweighting reflects the hidden interval structure of the environment.

To illustrate how to read these figures, consider the subplot labeled $K > 1.0$, progress $= 0.6$ in Figure 6. This subplot shows the weight vector output by the WeightNet trained in the standard Othello environment (Layout 1), evaluated at a game state where 60% of the board is occupied. The $x$-axis varies the agent's disc occupancy ratio $\rho$ from 0% to 100%, and the $y$-axis shows the resulting weight assigned to each of the four sub-utilities. In this case, corner and mobility receive consistently high positive weights across all $\rho$ values, indicating that the agent prioritizes positional control regardless of its current disc share.

Figure 6 and Figure 7 show the output **w** of WeightNet in layout 1 (Standard $8 \times 8$ without obstacles) and layout 2 (Obstacles on corners). The results indicate that WeightNet is trained differently for different winning conditions $K$. For instance, in layout 1, the $K > 1.0$ condition assigns high positive importance to corners and mobility, while PosScore remains relatively less important. By contrast, for $K < 0.0$, PosScore receives high positive importance, whereas discDiff and Corner tend to become negative, reflecting that lower occupancy is strategically favorable. These patterns suggest that the adaptive Minimax is able to internalize broad differences between standard and inverse objectives.

For narrow regimes, the relevant question is whether the learned weights become sensitive to $\rho$ in a way that reflects the need for terminal occupancy control. In some cases, such sensitivity is visible. For example, mobility increases with $\rho$ for $K = 0.2$ in Figure 6, and PosScore decreases with $\rho$ for $K = 0.6$ in Figure 7, suggesting that the agent captures at least part of the relationship between occupancy and the latent winning interval. However, many other narrow-regime cases remain comparatively flat, such as $K = 0.4$ in Figure 6 and $K = 0.2$ in Figure 7. This indicates that narrow-regime adaptation is not absent, but inconsistent. In some environments, the learned utility becomes meaningfully occupancy-sensitive, whereas in others, WeightNet fails to establish a useful relationship between $\rho$ and the hidden objective.

This interpretation is also consistent with the per-environment results in Table 2. Even in very narrow regimes, the adaptive Minimax sometimes achieves non-trivial performance in specific environments (e.g., in $K = 0.6$ with $L = 7$, the adaptive Minimax achieves a 45% win rate against MCTS-100), whereas in $L = 1$ with the same $K = 0.6$, it drops to 5%. Such variability suggests that the difficulty of narrow-regime adaptation is not captured well by a single average alone. One plausible explanation is the stochastic nature of survival-only evolution. Since selection occurs through pairwise survival rather than direct optimization toward a shaped objective, there is no guarantee that a partially successful WeightNet will survive across generations. Under a relatively small population size for a difficult latent-objective problem, adaptation can therefore depend substantially on evolutionary luck, which may explain why useful narrow-regime behavior appears only in some environments.

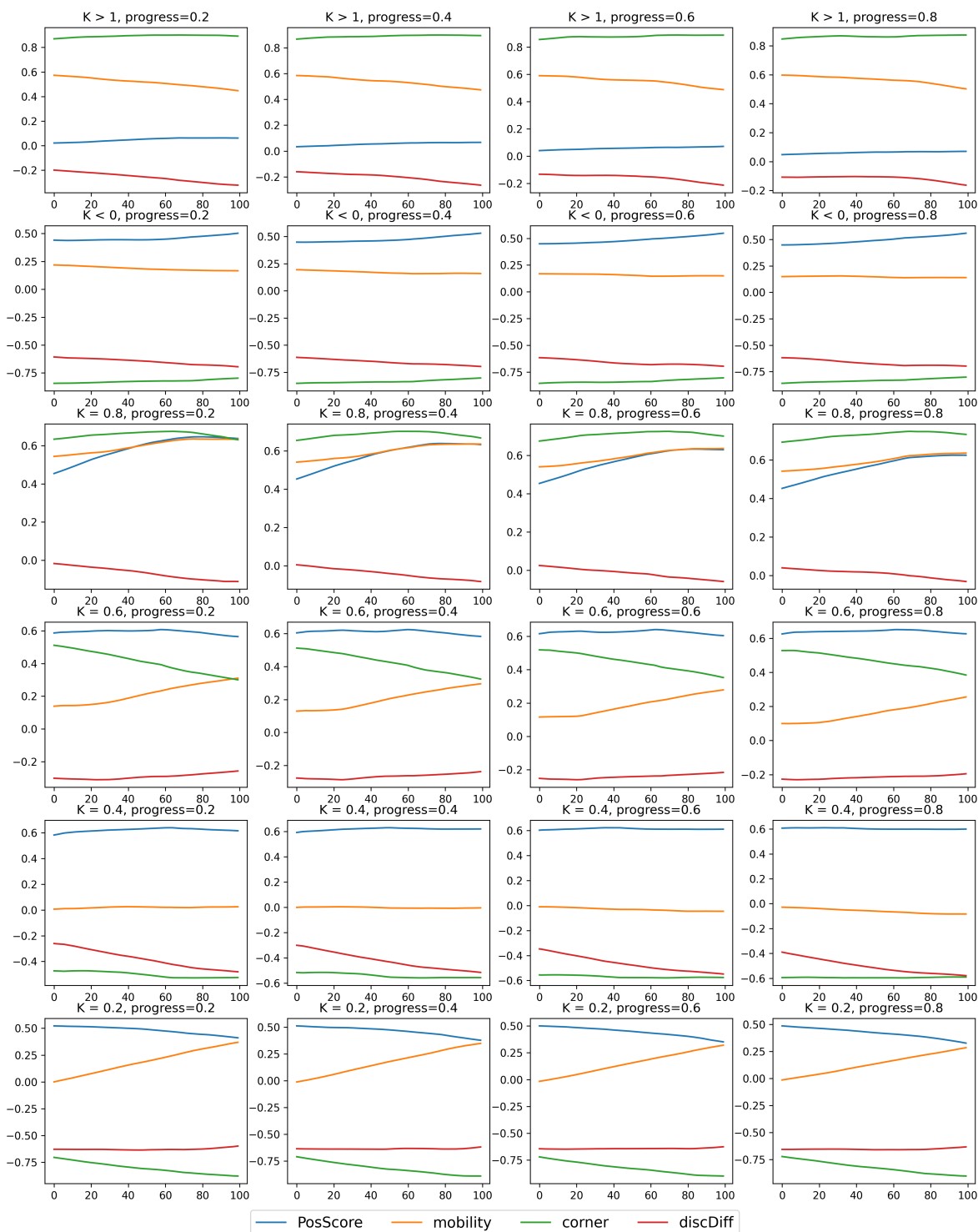

Figure 6: Output weights of the WeightNet with respect to player disc occupancy $\rho$, on various $K$ and progresses, in layout 1 (Standard $8 \times 8$ without obstacles)

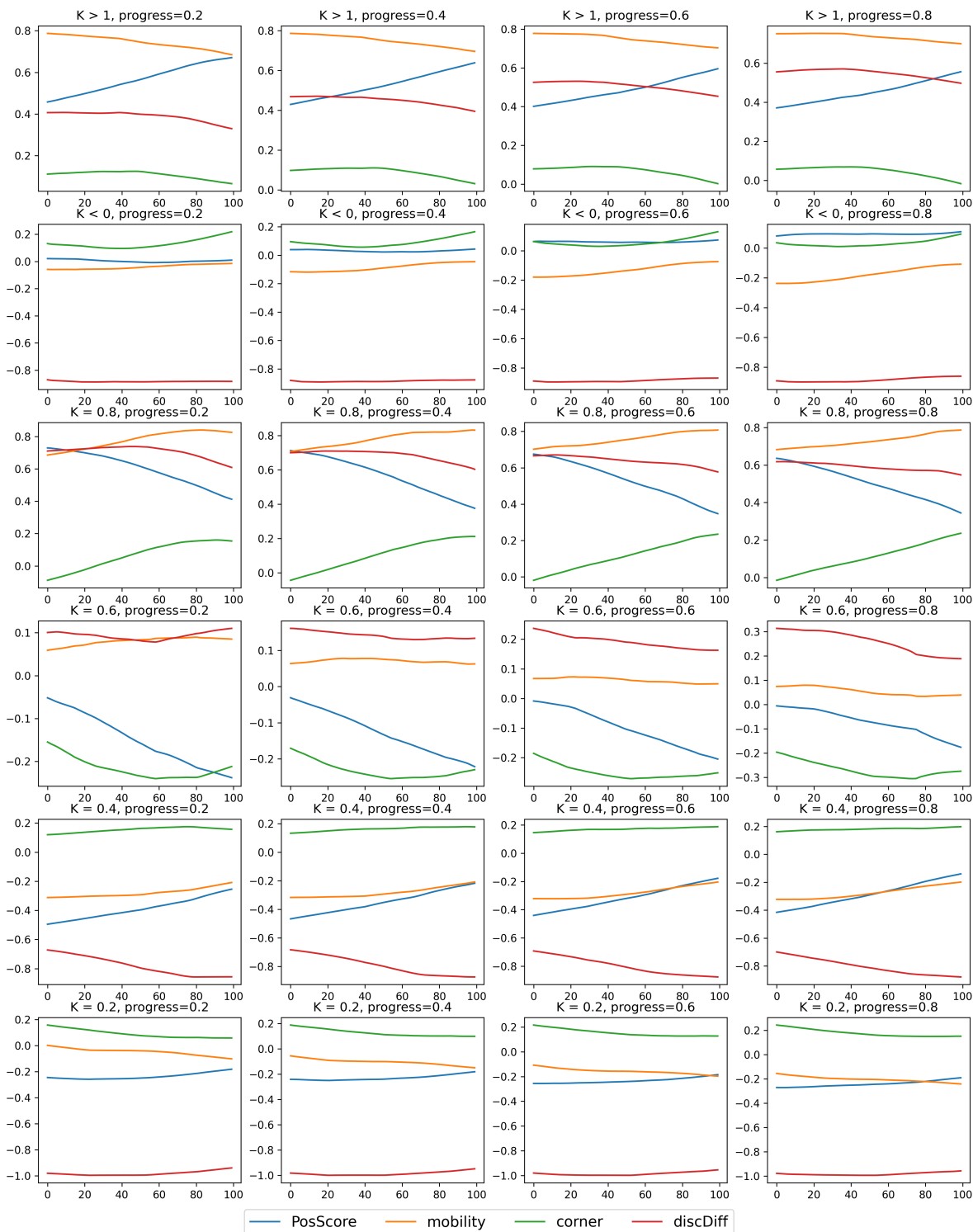

Figure 7: Output weights of the WeightNet with respect to player disc occupancy $\rho$, on various $K$ and progresses, in layout 2 (Obstacles on corners)

Table 2: Win/Draw/Lose rates (%) of the adaptive Minimax (2,000 games / 20 generations trained) against various baselines per environment. $C$ denotes the victory condition and $L$ denotes the board layout index (see Figure 1). [†]10-turn variants.

| $C$ | $L$ | Random | PosNet | PPO-2K | MCTS-30 | MCTS-50 | MCTS-100 |
|---|---|---|---|---|---|---|---|
| $K > 1.0$ | 1 | 100/0/0 | 80/5/15 | 100/0/0 | 90/0/10 | 95/0/5 | 100/0/0 |
| | 2 | 100/0/0 | 95/0/5 | 100/0/0 | 100/0/0 | 100/0/0 | 95/0/5 |
| | 3 | 95/0/5 | 100/0/0 | 100/0/0 | 70/5/25 | 90/0/10 | 85/0/15 |
| | 4 | 100/0/0 | 100/0/0 | 100/0/0 | 95/0/5 | 95/0/5 | 90/0/10 |
| | 5 | 100/0/0 | 25/0/75 | 100/0/0 | 80/0/20 | 85/5/10 | 60/0/40 |
| | 6 | 95/0/5 | 100/0/0 | 100/0/0 | 85/0/15 | 75/0/25 | 60/5/35 |
| | 7 | 80/10/10 | 50/0/50 | 0/0/100 | 75/0/25 | 50/10/40 | 60/0/40 |
| $K = 0.8$ | 1 | 75/25/0 | 55/30/15 | 50/50/0 | 75/25/0 | 60/35/5 | 70/25/5 |
| | 2 | 50/50/0 | 35/65/0 | 50/50/0 | 40/55/5 | 45/40/15 | 70/20/10 |
| | 3 | 50/40/10 | 70/30/0 | 100/0/0 | 40/50/10 | 60/15/25 | 40/30/30 |
| | 4 | 60/40/0 | 50/50/0 | 50/50/0 | 60/35/5 | 60/40/0 | 65/20/15 |
| | 5 | 75/10/15 | 75/0/25 | 50/0/50 | 40/5/55 | 15/0/85 | 40/0/60 |
| | 6 | 25/70/5 | 0/100/0 | 50/50/0 | 30/45/25 | 25/30/45 | 10/40/50 |
| | 7 | 45/15/40 | 50/0/50 | 100/0/0 | 45/5/50 | 45/10/45 | 15/5/80 |
| $K = 0.6$ | 1 | 40/55/5 | 15/75/10 | 0/100/0 | 15/80/5 | 15/85/0 | 5/95/0 |
| | 2 | 10/85/5 | 20/80/0 | 0/100/0 | 0/85/15 | 0/85/15 | 10/50/40 |
| | 3 | 10/90/0 | 0/100/0 | 0/100/0 | 5/75/20 | 5/75/20 | 10/75/15 |
| | 4 | 20/80/0 | 0/50/50 | 0/100/0 | 15/70/15 | 15/55/30 | 5/80/15 |
| | 5 | 25/75/0 | 0/100/0 | 0/100/0 | 10/65/25 | 0/80/20 | 5/80/15 |
| | 6 | 20/55/25 | 50/50/0 | 0/100/0 | 30/40/30 | 15/60/25 | 15/45/40 |
| | 7 | 10/75/15 | 50/50/0 | 0/50/50 | 25/55/20 | 15/70/15 | 45/50/5 |
| $K = 0.4$ | 1 | 0/100/0 | 15/85/0 | 0/100/0 | 0/100/0 | 5/90/5 | 0/100/0 |
| | 2 | 15/85/0 | 0/100/0 | 0/100/0 | 20/80/0 | 25/55/20 | 35/60/5 |
| | 3 | 25/75/0 | 0/100/0 | 50/50/0 | 10/85/5 | 10/75/15 | 10/70/20 |
| | 4 | 10/80/10 | 0/100/0 | 0/100/0 | 0/95/5 | 10/85/5 | 10/85/5 |
| | 5 | 20/60/20 | 10/85/5 | 50/50/0 | 0/30/70 | 0/25/75 | 5/35/60 |
| | 6 | 20/65/15 | 0/100/0 | 0/50/50 | 5/60/35 | 15/35/50 | 0/45/55 |
| | 7 | 40/45/15 | 0/100/0 | 0/100/0 | 15/40/45 | 10/15/75 | 0/25/75 |
| $K = 0.2$ | 1 | 70/30/0 | 40/60/0 | 0/100/0 | 50/45/5 | 40/50/10 | 60/40/0 |
| | 2 | 90/10/0 | 55/45/0 | 100/0/0 | 70/25/5 | 75/5/20 | 65/10/25 |
| | 3 | 85/10/5 | 75/25/0 | 100/0/0 | 85/0/15 | 60/10/30 | 60/15/25 |
| | 4 | 65/35/0 | 50/50/0 | 100/0/0 | 55/30/15 | 60/25/15 | 40/50/10 |
| | 5 | 95/0/5 | 75/25/0 | 0/0/100 | 60/0/40 | 30/0/70 | 30/15/55 |
| | 6 | 60/25/15 | 0/100/0 | 50/0/50 | 20/15/65 | 15/20/65 | 10/30/60 |
| | 7 | 80/5/15 | 100/0/0 | 100/0/0 | 85/0/15 | 60/15/25 | 50/10/40 |
| $K < 0.0$ | 1 | 100/0/0 | 100/0/0 | 100/0/0 | 100/0/0 | 95/0/5 | 95/0/5 |
| | 2 | 100/0/0 | 100/0/0 | 100/0/0 | 70/5/25 | 75/5/20 | 75/10/15 |
| | 3 | 95/0/5 | 100/0/0 | 100/0/0 | 80/0/20 | 50/25/25 | 40/10/50 |
| | 4 | 95/5/0 | 100/0/0 | 100/0/0 | 85/0/15 | 85/0/15 | 60/0/40 |
| | 5 | 85/10/5 | 100/0/0 | 100/0/0 | 30/0/70 | 15/0/85 | 20/0/80 |
| | 6 | 95/0/5 | 100/0/0 | 100/0/0 | 30/0/70 | 30/0/70 | 45/0/55 |
| | 7 | 90/0/10 | 100/0/0 | 50/0/50 | 65/5/30 | 65/10/25 | 55/5/40 |
| $K > 1.0$[†] | 1 | 95/0/5 | 95/5/0 | 100/0/0 | 90/0/10 | 90/0/10 | 90/0/10 |
| | 2 | 95/0/5 | 90/10/0 | 50/50/0 | 65/25/10 | 55/15/30 | 85/5/10 |
| | 3 | 95/0/5 | 100/0/0 | 100/0/0 | 40/45/15 | 55/25/20 | 45/40/15 |
| | 4 | 90/10/0 | 100/0/0 | 100/0/0 | 60/40/0 | 75/5/20 | 50/35/15 |
| | 5 | 90/5/5 | 100/0/0 | 100/0/0 | 75/15/10 | 45/25/30 | 60/20/20 |
| | 6 | 80/20/0 | 50/0/50 | 50/50/0 | 55/25/20 | 60/30/10 | 15/85/0 |
| | 7 | 85/10/5 | 50/50/0 | 100/0/0 | 70/20/10 | 70/25/5 | 50/40/10 |
| $K < 0.0$[†] | 1 | 100/0/0 | 100/0/0 | 50/50/0 | 100/0/0 | 90/5/5 | 95/0/5 |
| | 2 | 85/15/0 | 75/5/20 | 50/50/0 | 65/10/25 | 50/35/15 | 50/10/40 |
| | 3 | 85/10/5 | 50/50/0 | 100/0/0 | 60/20/20 | 55/15/30 | 50/0/50 |
| | 4 | 85/15/0 | 100/0/0 | 100/0/0 | 50/45/5 | 50/30/20 | 50/20/30 |
| | 5 | 80/20/0 | 100/0/0 | 100/0/0 | 55/30/15 | 60/10/30 | 50/25/25 |
| | 6 | 70/20/10 | 50/50/0 | 50/0/50 | 55/15/30 | 50/0/50 | 45/5/50 |
| | 7 | 85/5/10 | 100/0/0 | 100/0/0 | 45/20/35 | 50/25/25 | 50/15/35 |

# H   Detailed Results of Convergence Speed

In this section, we address the performance of adaptive Minimax and PPO agents against Random and MCTS-100 agents. Table 3 addresses the adaptive Minimax, and Table 4 addresses the PPO agent.

Table 3: Win/Draw/Lose rates (%) of the adaptive Minimax across different training generations (1, 5, 10, 20) against Random and MCTS-100 baselines. $C$ denotes the victory condition and $L$ denotes the board layout index. † denotes 10-turn variants. Note that each generation requires 100 games due to the population size of 50.

| $C$ | $L$ | Random | | | | MCTS-100 | | | |
|---|---|---|---|---|---|---|---|---|---|
| | | 1G | 5G | 10G | 20G | 1G | 5G | 10G | 20G |
| $K > 1.0$ | 1 | 95/0/5 | 100/0/0 | 100/0/0 | 100/0/0 | 80/0/20 | 100/0/0 | 90/10/0 | 100/0/0 |
| | 2 | 65/0/35 | 95/5/0 | 100/0/0 | 100/0/0 | 45/0/55 | 95/0/5 | 65/0/35 | 95/0/5 |
| | 3 | 100/0/0 | 95/0/5 | 100/0/0 | 95/0/5 | 90/0/10 | 60/0/40 | 85/0/15 | 85/0/15 |
| | 4 | 100/0/0 | 100/0/0 | 100/0/0 | 100/0/0 | 90/0/10 | 100/0/0 | 100/0/0 | 90/0/10 |
| | 5 | 95/0/5 | 95/0/5 | 100/0/0 | 100/0/0 | 65/0/35 | 85/5/10 | 85/5/10 | 60/0/40 |
| | 6 | 85/5/10 | 80/0/20 | 80/0/20 | 95/0/5 | 40/0/60 | 40/0/60 | 65/5/30 | 60/5/35 |
| | 7 | 65/0/35 | 95/0/5 | 95/0/5 | 80/10/10 | 25/5/70 | 85/0/15 | 90/0/10 | 60/0/40 |
| $K = 0.8$ | 1 | 65/5/30 | 60/40/0 | 70/30/0 | 75/25/0 | 5/0/95 | 55/45/0 | 75/25/0 | 70/25/5 |
| | 2 | 40/10/50 | 60/40/0 | 35/65/0 | 50/50/0 | 0/0/100 | 15/50/35 | 45/40/15 | 70/20/10 |
| | 3 | 50/20/30 | 20/70/10 | 50/50/0 | 50/40/10 | 0/0/100 | 50/50/0 | 30/65/5 | 40/30/30 |
| | 4 | 30/70/0 | 35/65/0 | 70/25/5 | 60/40/0 | 15/65/20 | 45/55/0 | 45/50/5 | 65/20/15 |
| | 5 | 55/5/40 | 80/15/5 | 90/0/10 | 75/10/15 | 40/0/60 | 50/30/20 | 40/5/55 | 40/0/60 |
| | 6 | 10/70/20 | 30/65/5 | 40/60/0 | 25/70/5 | 20/50/30 | 10/60/30 | 15/55/30 | 10/40/50 |
| | 7 | 50/15/35 | 80/10/10 | 85/10/5 | 45/15/40 | 5/10/85 | 75/10/15 | 60/0/40 | 15/5/80 |
| $K = 0.6$ | 1 | 35/45/20 | 30/60/10 | 30/70/0 | 40/55/5 | 30/20/50 | 15/55/30 | 15/60/25 | 5/95/0 |
| | 2 | 5/60/35 | 10/90/0 | 10/85/5 | 10/85/5 | 10/45/45 | 10/70/20 | 15/70/15 | 10/50/40 |
| | 3 | 25/50/25 | 15/70/15 | 10/85/5 | 10/90/0 | 5/65/30 | 10/45/45 | 0/40/60 | 10/75/15 |
| | 4 | 10/80/10 | 10/85/5 | 15/70/15 | 20/80/0 | 5/65/30 | 10/65/25 | 10/65/25 | 5/80/15 |
| | 5 | 10/80/10 | 30/45/25 | 10/90/0 | 25/75/0 | 5/60/35 | 5/5/90 | 5/60/35 | 5/80/15 |
| | 6 | 15/85/0 | 15/70/15 | 15/60/25 | 20/55/25 | 0/90/10 | 15/55/30 | 15/55/30 | 15/45/40 |
| | 7 | 60/40/0 | 25/70/5 | 5/85/10 | 10/75/15 | 40/35/25 | 5/80/15 | 10/65/25 | 45/50/5 |
| $K = 0.4$ | 1 | 10/75/15 | 35/45/20 | 5/95/0 | 0/100/0 | 5/45/50 | 0/55/45 | 10/90/0 | 0/100/0 |
| | 2 | 20/80/0 | 5/65/30 | 35/60/5 | 15/85/0 | 15/75/10 | 0/50/50 | 15/45/40 | 35/60/5 |
| | 3 | 10/55/35 | 35/65/0 | 25/70/5 | 25/75/0 | 0/50/50 | 5/65/30 | 25/50/25 | 10/70/20 |
| | 4 | 15/80/5 | 20/75/5 | 0/100/0 | 10/80/10 | 0/30/70 | 10/20/70 | 10/80/10 | 10/85/5 |
| | 5 | 25/50/25 | 30/55/15 | 25/40/35 | 20/60/20 | 15/25/60 | 20/35/45 | 0/15/85 | 5/35/60 |
| | 6 | 30/70/0 | 10/85/5 | 15/65/20 | 20/65/15 | 15/60/25 | 10/55/35 | 0/50/50 | 0/45/55 |
| | 7 | 45/40/15 | 20/70/10 | 25/50/25 | 40/45/15 | 0/30/70 | 10/20/70 | 10/10/80 | 0/25/75 |
| $K = 0.2$ | 1 | 90/5/5 | 60/40/0 | 80/20/0 | 70/30/0 | 50/10/40 | 60/35/5 | 35/45/20 | 60/40/0 |
| | 2 | 40/10/50 | 80/10/10 | 80/10/10 | 90/10/0 | 0/5/95 | 45/10/45 | 65/10/25 | 65/10/25 |
| | 3 | 25/5/70 | 90/10/0 | 90/10/0 | 85/10/5 | 0/30/70 | 50/0/50 | 45/10/45 | 60/15/25 |
| | 4 | 5/20/75 | 85/10/5 | 70/25/5 | 65/35/0 | 5/0/95 | 35/0/65 | 40/40/20 | 40/50/10 |
| | 5 | 60/10/30 | 85/10/5 | 65/5/30 | 95/0/5 | 10/5/85 | 0/5/95 | 20/0/80 | 30/15/55 |
| | 6 | 45/40/15 | 35/25/40 | 50/30/20 | 60/25/15 | 20/10/70 | 10/10/80 | 25/10/65 | 10/30/60 |
| | 7 | 25/10/65 | 85/5/10 | 85/10/5 | 80/5/15 | 0/0/100 | 15/5/80 | 40/0/60 | 50/10/40 |
| $K < 0.0$ | 1 | 35/0/65 | 100/0/0 | 100/0/0 | 100/0/0 | 5/0/95 | 85/5/10 | 90/0/10 | 95/0/5 |
| | 2 | 5/0/95 | 85/0/15 | 95/0/5 | 100/0/0 | 0/0/100 | 30/10/60 | 70/0/30 | 75/10/15 |
| | 3 | 5/0/95 | 100/0/0 | 90/0/10 | 95/0/5 | 0/0/100 | 35/15/50 | 60/10/30 | 40/10/50 |
| | 4 | 85/0/15 | 95/0/5 | 100/0/0 | 95/5/0 | 5/0/95 | 50/0/50 | 80/0/20 | 60/0/40 |
| | 5 | 65/0/35 | 65/5/30 | 90/0/10 | 85/10/5 | 5/0/95 | 5/0/95 | 20/0/80 | 20/0/80 |
| | 6 | 70/5/25 | 75/0/25 | 65/0/35 | 95/0/5 | 25/0/75 | 20/5/75 | 15/0/85 | 45/0/55 |
| | 7 | 90/5/5 | 70/5/25 | 80/0/20 | 90/0/10 | 15/10/75 | 5/5/90 | 20/10/70 | 55/5/40 |
| $K > 1.0^\dagger$ | 1 | 90/0/10 | 100/0/0 | 90/5/5 | 95/0/5 | 50/5/45 | 95/0/5 | 90/0/10 | 90/0/10 |
| | 2 | 30/10/60 | 85/15/0 | 85/10/5 | 95/0/5 | 0/0/100 | 75/25/0 | 60/20/20 | 85/5/10 |
| | 3 | 5/15/80 | 80/20/0 | 100/0/0 | 95/0/5 | 0/5/95 | 65/10/25 | 45/25/30 | 45/40/15 |
| | 4 | 70/20/10 | 90/5/5 | 75/20/5 | 90/10/0 | 50/5/45 | 75/5/20 | 70/10/20 | 50/35/15 |
| | 5 | 85/15/0 | 100/0/0 | 100/0/0 | 90/5/5 | 35/30/35 | 45/40/15 | 20/45/35 | 60/20/20 |
| | 6 | 75/5/20 | 75/20/5 | 90/10/0 | 80/20/0 | 10/35/55 | 5/75/20 | 30/70/0 | 15/85/0 |

| C | L | Random | | | | MCTS-100 | | | |
|---|---|---|---|---|---|---|---|---|---|
| | | 1G | 5G | 10G | 20G | 1G | 5G | 10G | 20G |
| | 7 | 85/10/5 | 85/0/15 | 85/5/10 | 85/10/5 | 45/15/40 | 40/20/40 | 60/25/15 | 50/40/10 |
| $K < 0.0^\dagger$ | 1 | 40/5/55 | 100/0/0 | 100/0/0 | 100/0/0 | 0/0/100 | 80/0/20 | 90/5/5 | 95/0/5 |
| | 2 | 5/20/75 | 80/20/0 | 85/15/0 | 85/15/0 | 0/0/100 | 55/5/40 | 50/5/45 | 50/10/40 |
| | 3 | 0/25/75 | 85/10/5 | 65/30/5 | 85/10/5 | 0/0/100 | 45/15/40 | 55/5/40 | 50/0/50 |
| | 4 | 70/30/0 | 70/15/15 | 75/25/0 | 85/15/0 | 45/20/35 | 50/10/40 | 50/10/40 | 50/20/30 |
| | 5 | 75/20/5 | 75/20/5 | 90/5/5 | 80/20/0 | 50/20/30 | 50/15/35 | 50/20/30 | 50/25/25 |
| | 6 | 70/20/10 | 70/5/25 | 85/10/5 | 70/20/10 | 60/10/30 | 60/0/40 | 5/45/50 | 45/5/50 |
| | 7 | 65/15/20 | 90/5/5 | 80/15/5 | 85/5/10 | 55/10/35 | 55/20/25 | 50/10/40 | 50/15/35 |

Table 4: Win/Draw/Lose rates (%) of PPO across different training episodes (200, 2000, 10000) against Random and MCTS-100 baselines. $C$ denotes the victory condition and $L$ denotes the board layout index. † denotes 10-turn variants.

| C | L | Random | | | MCTS-100 | | |
|---|---|---|---|---|---|---|---|
| | | 200 EP | 2K EP | 10K EP | 200 EP | 2K EP | 10K EP |
| $K > 1.0$ | 1 | 70/0/30 | 90/5/5 | 65/5/30 | 20/5/75 | 10/5/85 | 5/0/95 |
| | 2 | 50/10/40 | 75/10/15 | 80/0/20 | 0/0/100 | 10/0/90 | 10/5/85 |
| | 3 | 45/15/40 | 70/0/30 | 70/0/30 | 15/0/85 | 0/10/90 | 0/5/95 |
| | 4 | 55/0/45 | 80/5/15 | 80/0/20 | 5/0/95 | 25/0/75 | 20/0/80 |
| | 5 | 50/5/45 | 80/0/20 | 95/0/5 | 10/0/90 | 30/0/70 | 35/0/65 |
| | 6 | 35/0/65 | 85/0/15 | 80/0/20 | 10/0/90 | 20/0/80 | 35/0/65 |
| | 7 | 60/5/35 | 70/0/30 | 90/5/5 | 0/0/100 | 25/0/75 | 25/0/75 |
| $K = 0.8$ | 1 | 60/15/25 | 70/20/10 | 80/5/15 | 0/0/100 | 10/5/85 | 10/15/75 |
| | 2 | 75/5/20 | 80/0/20 | 60/10/30 | 15/0/85 | 15/0/85 | 5/10/85 |
| | 3 | 75/5/20 | 75/5/20 | 60/15/25 | 10/0/90 | 10/10/80 | 5/5/90 |
| | 4 | 45/20/35 | 75/0/25 | 65/5/30 | 5/0/95 | 20/0/80 | 20/0/80 |
| | 5 | 70/0/30 | 65/10/25 | 80/5/15 | 15/0/85 | 25/5/70 | 45/0/55 |
| | 6 | 45/15/40 | 35/30/35 | 55/20/25 | 5/5/90 | 15/15/70 | 20/10/70 |
| | 7 | 60/5/35 | 80/5/15 | 90/0/10 | 0/0/100 | 15/0/85 | 20/5/75 |
| $K = 0.6$ | 1 | 30/35/35 | 25/50/25 | 20/60/20 | 15/20/65 | 5/30/65 | 0/20/80 |
| | 2 | 25/55/20 | 25/40/35 | 25/55/20 | 15/20/65 | 10/10/80 | 5/30/65 |
| | 3 | 35/35/30 | 30/50/20 | 35/55/10 | 5/30/65 | 0/20/80 | 15/25/60 |
| | 4 | 45/35/20 | 25/55/20 | 35/50/15 | 5/40/55 | 0/35/65 | 5/35/60 |
| | 5 | 20/50/30 | 25/40/35 | 40/40/20 | 5/15/80 | 5/25/70 | 10/10/80 |
| | 6 | 15/65/20 | 10/75/15 | 20/70/10 | 0/55/45 | 0/35/65 | 5/25/70 |
| | 7 | 15/65/20 | 25/60/15 | 50/30/20 | 5/10/85 | 0/10/90 | 10/15/75 |
| $K = 0.4$ | 1 | 25/50/25 | 45/50/5 | 20/55/25 | 10/25/65 | 0/20/80 | 10/25/65 |
| | 2 | 30/45/25 | 40/55/5 | 30/40/30 | 10/30/60 | 0/25/75 | 5/50/45 |
| | 3 | 20/45/35 | 30/60/10 | 40/45/15 | 0/35/65 | 0/40/60 | 0/25/75 |
| | 4 | 30/55/15 | 10/50/40 | 15/65/20 | 15/25/60 | 15/45/40 | 0/30/70 |
| | 5 | 30/50/20 | 25/55/20 | 15/80/5 | 0/30/70 | 5/10/85 | 0/25/75 |
| | 6 | 10/75/15 | 0/95/5 | 10/75/15 | 0/65/35 | 5/35/60 | 0/25/75 |
| | 7 | 50/20/30 | 35/40/25 | 50/25/25 | 5/0/95 | 0/25/75 | 5/5/90 |
| $K = 0.2$ | 1 | 65/0/35 | 60/0/40 | 60/15/25 | 20/5/75 | 10/0/90 | 0/10/90 |
| | 2 | 35/5/60 | 85/5/10 | 50/10/40 | 0/5/95 | 15/0/85 | 5/0/95 |
| | 3 | 55/25/20 | 75/10/15 | 75/15/10 | 5/0/95 | 0/0/100 | 15/0/85 |
| | 4 | 40/10/50 | 70/5/25 | 55/10/35 | 0/5/95 | 5/0/95 | 0/0/100 |
| | 5 | 75/0/25 | 75/5/20 | 80/5/15 | 0/0/100 | 0/5/95 | 0/0/100 |
| | 6 | 35/45/20 | 45/35/20 | 40/35/25 | 10/15/75 | 10/20/70 | 5/20/75 |
| | 7 | 65/10/25 | 85/5/10 | 70/0/30 | 5/0/95 | 5/0/95 | 5/5/90 |
| $K < 0.0$ | 1 | 80/0/20 | 90/5/5 | 95/0/5 | 0/0/100 | 0/0/100 | 5/0/95 |
| | 2 | 45/20/35 | 70/0/30 | 70/0/30 | 0/0/100 | 5/0/95 | 0/0/100 |
| | 3 | 60/20/20 | 85/0/15 | 85/0/15 | 0/0/100 | 0/0/100 | 15/10/75 |
| | 4 | 75/5/20 | 80/0/20 | 90/0/10 | 10/0/90 | 10/5/85 | 5/0/95 |
| | 5 | 70/0/30 | 65/10/25 | 80/0/20 | 0/5/95 | 0/0/100 | 0/0/100 |
| | 6 | 55/0/45 | 50/0/50 | 75/0/25 | 10/0/90 | 25/0/75 | 40/0/60 |
| | 7 | 65/10/25 | 85/5/10 | 85/10/5 | 10/0/90 | 0/0/100 | 15/10/75 |

| $C$ | $L$ | Random | | | MCTS-100 | | |
|---|---|---|---|---|---|---|---|
| | | 200 EP | 2K EP | 10K EP | 200 EP | 2K EP | 10K EP |
| $K > 1.0^{\dagger}$ | 1 | 70/0/30 | 90/5/5 | 65/5/30 | 20/5/75 | 10/5/85 | 5/0/95 |
| | 2 | 35/30/35 | 60/25/15 | 55/30/15 | 0/0/100 | 20/15/65 | 50/0/50 |
| | 3 | 30/45/25 | 40/35/25 | 75/10/15 | 0/10/90 | 35/5/60 | 10/50/40 |
| | 4 | 40/25/35 | 65/10/25 | 70/20/10 | 15/5/80 | 15/5/80 | 45/10/45 |
| | 5 | 45/15/40 | 45/20/35 | 65/20/15 | 10/0/90 | 15/5/80 | 35/20/45 |
| | 6 | 45/30/25 | 90/5/5 | 75/20/5 | 0/5/95 | 5/50/45 | 10/50/40 |
| | 7 | 35/25/40 | 40/40/20 | 65/20/15 | 5/20/75 | 15/20/65 | 15/5/80 |
| $K < 0.0^{\dagger}$ | 1 | 80/0/20 | 90/5/5 | 95/0/5 | 0/0/100 | 0/0/100 | 5/0/95 |
| | 2 | 35/30/35 | 50/20/30 | 75/15/10 | 0/0/100 | 5/25/70 | 20/20/60 |
| | 3 | 25/45/30 | 60/25/15 | 70/0/30 | 0/5/95 | 50/5/45 | 30/20/50 |
| | 4 | 35/25/40 | 60/25/15 | 75/0/25 | 10/5/85 | 10/20/70 | 25/25/50 |
| | 5 | 40/15/45 | 65/20/15 | 70/20/10 | 0/5/95 | 35/20/45 | 30/20/50 |
| | 6 | 25/30/45 | 65/15/20 | 60/5/35 | 0/5/95 | 50/0/50 | 50/5/45 |
| | 7 | 40/25/35 | 70/20/10 | 75/15/10 | 5/10/85 | 25/0/75 | 40/10/50 |

