# OpenReview forum: "The Expanded Othello AI Arena: Evaluating Intelligent Systems Through Constrained Adaptation to Unseen Conditions"
_TMLR — Accepted by TMLR_

### Review · Reviewer_gHtG · 2026-03-02

**Summary Of Contributions:**

**Summary**:

This submission presents a new benchmark that evaluates meta-learning and sample efficiency in a partially-observable two player game, namely an extended version of Othello. Othello traditionally involves placing discs on a square grid, in order to sandwich and flip the opponent's discs and maximize the number of discs controlled. This benchmark proposes a two-fold extension by considering diverse 7 grid morphologies (observable by the agent) and a continuous hidden disc threshold value, which determines the objective of the game. The second modification is particularly interesting, as it induces dynamic changes in players' objectives. By discretizing the threshold in 6 levels, the benchmark proposes 54 different configurations. On each configuration, agents are allowed to train for 2000 games in order to highlight sample efficient learning instead of asymptotic performance.
The authors evaluate a range of approaches over the benchmark, namely (i) a min-max search agent optimizing meta-learned evaluations of the grid and a learned evaluation function, (ii) a MCTS agent with sample-based access to the hidden objective, (iii) PPO as a model-free RL algorithm. While (ii) represents a very strong baseline for large search budgets, (i) and (iii) are presented as realistic approaches for the benchmark. While they perform well for the standard max-occupancy objective, these approaches underperform for very strict constraints on the disc-ratio.

**Contributions:**

This work has two main contributions: the introduction of a benchmark, and an empirical evaluation of baselines on it. The former is particularly valuable, as the proposed benchmark stresses inference of hidden game rules and anticipation of the opponent's behavior, which is not common in existing benchmarks.

**Strengths:**
- This work is very well written and cleanly explained.
- The clean separation of seen and unseen environment characteristic allows evaluation of meta-learning algorithm across two important axes of generalization.
- While I am not entirely familiar with the literature, I am not aware of an existing overlapping benchmark.

**Weakesses:**

- Empirical results do not entirely support claims on what the benchmark should be able to capture. Please, see the next paragraph for details.
- The benchmark is, to the best of my knowledge, somewhat underspecified. The authors should define clear rules for evaluation on the game. Namely, is access to observable dynamics (i.e. where discs can be placed and what changes their placement induces) allowed? Over the 2000 games allowed, can agents perform self-play? Or is their opponent fixed? Clearly defining an evaluation protocol in terms of what is allowed exactly would be beneficial (e.g. the simulator can be queried sequentially for 2000 games, and the agent can control both players).
- The baselines considered are not exactly uniform or comparable. MCTS is privileged due to its access to the hidden objective through sampling, and its budget can be bounded arbitrarily. PPO uses 10K games for training of 2K, but has no knowledge of dynamics. To the best of my understanding, the Minimax agent leverages dynamic information, trains its WeightNet for 2K games, but its PosNet for 20K games. Due to this differences, the behaviors of baselines are hard to compare, and mostly predefined by their access to more/less information/compute. Coupled with the above point, this makes the actual evaluation protocol the authors want to propose a little unclear.

**Other:**
- The authors suggest that this benchmark is suitable for evaluation of LLMs. If this is the case, an exact evaluation framework including prompts and basic inference code could be very helpful.
- I would encourage the authors on which real-world applications this benchmark could be shedding some light on.
- While "disc-ratio" is a good term, it can be confusing for those who have not played Othello. It could be explained in the introduction.
- I would encourage the authors to refer to established meta-learning environments beside zero-sum games and LLM-centric frameworks (see Section 5 of [1]).
- I am not entirely sure I understand the semantics of Figure 1. What do the colors represent? How are regime positions guessed?
- The confidence intervals of the top part of Table 1 seem to capture variations across different Ks. This results in very large intervals. Is this intended?
- Figure 3 is missing confidence intervals.

**References:**

[1] Nikulin et al., XLand-MiniGrid: Scalable Meta-Reinforcement Learning Environments in JAX, NeurIPS 2024 D&B

**Audience:**

Yes

**Audience Explanation:**

I an unfortunately not entirely familiar with the literature on benchmarks for meta-learning and zero-sum games. At the same time, I have not found any work directly overlapping with this one; I thus believe that this work will be of interest to the community.

**Broader Impact Concerns:**

I do not have concerns on ethical implications.

**Claims And Evidence:**

No

**Claims Explanation:**

My main concern with this submission is centered on the mismatch to what the benchmark is supposed to evaluate, and what it seems to actually evaluate according to empirical results.
Crucially, authors argue that doing well on the benchmark requires “meta-cognitive strategy synthesis”. However, a clean comparison between two instances of the same method (only one employing a meta-learned component) would be needed to check that meta-learning is indeed key for this problem. Furthermore, authors claim that mastering extended Othello requires "employing a primitive form of Theory of Mind (ToM)". This claim seems to be a stretch, as the better performing baselines arguably does not possess any ToM.

A second concern regard the numerical results. The random baseline appears to beat the Minimax strategy for $K=0.4$ and $K=0.6$. Why is this the case? I understand that the game becomes more nuanced in this setting, but an optimization approach should not actively decrease the winrate to the best of my knowledge.

**Requested Changes:**

I would suggest that the author:
- Explain, reformulate or support  the claims on what the benchmark is supposed to measure.
- Explain the counterintuitive performance of the Minimax agent under strong constraints.
- Clarify exactly the intended evaluation setting for this benchmark.
- Provide an evaluation of baseline within the constraints at the previous points, such that their performance may be comparable.

---

> ### Author Response · Authors · 2026-04-03
> **Response to reviewer gHtG**
>
> We thank the reviewer for the careful reading and constructive feedback. We revised the manuscript substantially in response to the main concerns.
>
> **1. Clarifying the Evaluation Setting**
>
> We agree that the previous manuscript did not specify the evaluation protocol clearly enough. We addressed this by revising **§2.4 Interaction Protocol** and related baseline descriptions.
>
> In §2.4, we now explicitly state that the agent is limited to **full sequential game interaction** within the budget, that **self-play is allowed**, and that **no additional simulator access** is permitted beyond such interaction. We also clarify that the protocol does not allow forward queries, branching rollouts, counterfactual state evaluation, or access to a separately specified external opponent policy.
>
> Related to this, we clarified the distinction between baselines in §4.4 and Appendix E, F. In the revised manuscript, **MCTS** is explicitly described as a **simulator-privileged reference**, rather than a matched-information adaptive baseline, whereas Adaptive Minimax and PPO are evaluated under the sequential interaction protocol without such privileged search-time access.
>
> **2. Clarifying What the Benchmark is Intended to Measure**
>
> We agree that the original manuscript made the ToM claim too strong, and we revised it substantially.
>
> This revision has two parts. First, in §2.3.2, we changed the narrow-regime environment semantics themselves. The original version explicitly included **opponent-forfeit-based victories**, which naturally encouraged a ToM-centered interpretation. In the revised version, we removed that structure and replaced it with a **draw-based interval rule**. We also introduced the discussion of *strategic withdrawal*, which better matches the revised environment. In narrow regimes, agents may face situations where reliably satisfying the win condition is difficult, while reaching a draw remains feasible.
>
> Second, we revised the paper’s framing and interpretation to match the new environment. In §1, we removed the stronger claim that successful adaptation requires ToM, and instead reframed the benchmark as one for **adaptation under latent objectives in adversarial play**. In §5.2, we no longer explain narrow-regime difficulty mainly through the lack of ToM. Instead, the revised discussion emphasizes that under the draw-based rule, low win rates can coexist with high draw rates, meaning that agents may still display meaningful non-losing behavior even when they do not reliably convert to wins. In §6, ToM is discussed only as a possible future direction rather than a demonstrated requirement.
>
> **3. Baseline Comparability and Ablation Concerns**
>
> We agree that baseline comparability needed to be clearer, and we revised both the descriptions and the PPO setup.
>
> We agree that cleaner ablations would be valuable in principle. However, a direct *Meta-RL*-style comparison is difficult here because the benchmark uses **variable board geometries with varying sizes**, so it is difficult to implement in the Expanded Othello setups. Likewise, removing **PosNet** from Adaptive Minimax would not isolate *meta-learning* alone; it would also remove the component that provides basic spatial structure across diverse layouts.
>
> We also substantially revised **PPO**. In the original version, PPO used a materially weaker setup, including a simpler training protocol against the Random agent. In the revised manuscript (§4.4, Appendix F), PPO has been redesigned to better match the benchmark setting: it is now trained under the same *2,000-game interaction budget* as Adaptive Minimax, uses **self-play** instead of a simpler fixed-opponent setup with legal-action masking via masked PPO. We also evaluated PPO against Random and MCTS-100 under increasing training budgets in **Figure 2**, and reported the results of Adaptive Minimax against PPO-2000 in Table 1. As a result, the revised PPO is substantially stronger than in the previous version, especially against the Random agent, while still remaining weaker than Adaptive Minimax in most environments. This makes PPO a much stronger and more appropriate baseline than in the original submission.
>
> **4. Clarifying the Numerical Results**
>
> We agree that some parts of the original results presentation, especially in narrow regimes, were not sufficiently clear. To address this, we revised Table 1, expanded the discussion in §5.2, and added Appendix Tables 2 and 3.
>
> The revised Table 1 now reports *Win/Draw/Lose* rates aggregated only across environments sharing the same victory condition, while the appendix provides *per-environment results*. Note that the standard deviation comes from the aggregation over environments sharing the same victory condition $C$. In the previous manuscript, Table 1 also reported aggregation over environments sharing $L$, which has a much higher standard deviation due to having heterogeneous $C$ environments.

---

> > ### Comment · Reviewer_gHtG · 2026-04-03
> >
> > Thank you for considering my comments and weakening the claims appropriately. I have no further questions.

---

### Review · Reviewer_Azjt · 2026-03-15

**Summary Of Contributions:**

This paper introduces a new environment for benchmarking AI systems based on Othello, with a particular focus on efficiently learning to adapt under incomplete information about the win condition. Due to the incomplete information, the systems need theory of mind capabilities to infer the missing information based on the behaviour of the opponent. The paper then provides some

## Strengths
- Expanded Othello AI Arena seems well-designed and likely to provide utility to the research community
- Clearly written
- Existing experiments have sensible design

## Weaknesses
- The budget of 2,000 episodes may be too generous to really test strategic information-seeking behaviours
- The PPO baseline may be too weak.

**Audience:**

Yes

**Audience Explanation:**

This paper introduces a new benchmark environment that focuses on Theory of Mind based reasoning, which is an aspect of intelligence often missing from other benchmarks.

**Claims And Evidence:**

No

**Claims Explanation:**

The proposed environment is clearly described, and code is provided.

I am on the fence regarding the utility of Expanded Othello AI Arena as proposed for testing theory of mind capabilities discussed in section 2. I think the existing experiments are sensibly designed given the proposed structure (though not sure why the confidence bounds in table 1 are so large). However, 2,000 episodes is quite a lot of time for adaptation if the purpose is to check whether the agents are capable of reasoning about the strategy of opponents. For this reason I will answer 'no' currently, though a productive discussion phase could change my mind.

Consider the following paragraphs from sec 2.3.2

> Consequently, the Expanded Othello environment is transformed from a stationary optimization task into
a rigorous test of online system identification and adversarial reasoning. This transition necessitates the
functional integration of Theory of Mind—the capacity to model and anticipate an opponent’s strategic
intentions—into the process of environmental adaptation. Because the winning condition C remains latent,
the agent must not only decipher the hidden success criteria from sparse terminal feedback but also reason
about the adversary’s perspective to secure the victory.

> For instance, consider a majority-constraint regime with K = 0.8. In the mid-game phase, consider two
players i and j, having disc occupancy ratios as ρi = 0.3 and ρj = 0.7. Low occupancy player i may conclude
that reaching the target winning range (0.5 < ρi < 0.8) is mathematically improbable through standard play.
Instead, they might pivot to a strategy of strategic concession, attempting to induce the opponent into a
forfeit by forcing them to exceed the K = 0.8 threshold. Conversely, the opponent j, anticipating this attempt
to engineer an over-domination, should adapt by safely reducing their own occupancy to remain within the
permissible bounds. This recursive strategic interplay demonstrates that successful adaptation in the Arena
requires deciphering not just the environment’s rules, but the opponent’s reactive behavior.

Based on these paragraphs, it seems that in order to really test "online system identification and adversarial reasoning", the adaptation budget should be extremely low (e.g. within a single game, or perhaps across a handful of games), particularly given that there's functionally one hidden parameter (K) to infer/adapt to.

One alternative evaluation protocol would be to report win rates after different amounts of interaction. E.g. either define some different timescale challenges (1, 10, 2000), or report metrics at regular intervals up to 2,000.

**Requested Changes:**

## Critical
1. Please see my comments re. the 2,000 game limit. Perhaps some extra text is needed to justify this further (though happy to discuss during the discussion phase).
2. Please elaborate on the confidence bounds in Table 1 -- why are they so large?
3. For games related to ToM please add some discussion of the following relevant citation: [1] N. Bard et al., ‘The Hanabi challenge: A new frontier for AI research’, Artificial Intelligence, vol. 280, p. 103216, Mar. 2020, doi: 10.1016/j.artint.2019.103216.

## Strengthening
3. The PPO baselines are probably too weak because they train a newly-initialised network for each task, and do not get any prior knowledge (the adaptive minimax agents get PosNet). Alternative RL-based schemes could be explored, but it would be reasonable to leave these to future work.

---

> ### Author Response · Authors · 2026-04-03
> **Response to reviewer Azjt**
>
> We thank the reviewer for the thoughtful and constructive feedback. We address the main concerns below:
>
> **1. Interaction Budget and Convergence Speed**
>
> We found the following suggestion particularly helpful:
> > One alternative evaluation protocol would be to report win rates after different amounts of interaction. [...]
>
> Following this suggestion, we revised the manuscript so that the evaluation is no longer presented only through the final result after 2,000 games. In §5.2, we now explicitly discuss adaptation **over training progress** of Adaptive Minimax and PPO against Random and MCTS-100, by adding **Figure 2** to show win rates under increasing training budgets. This approach well suits the concept of *skill-acquisition efficiency* we have claimed, by showing the **convergence speed** of the agents.
>
> This revision directly addresses the reviewer’s concern by making the **speed of convergence** visible, rather than relying only on final performance after 2,000 games. In particular, the revised manuscript now shows that the Adaptive Minimax often improves substantially **very early** (1st generation) in training, while PPO generally improves more gradually. Note that due to the evolution-based nature of Adaptive Minimax, training below one generation update (100 games) was impossible.
>
> **2. ToM Interpretation**
>
> We agree that the original ToM framing was too strong, and we revised it substantially.
>
> The revision has two parts. First, in §2.3.2, we changed the narrow-regime environment semantics themselves. The original version explicitly included **opponent-forfeit-based** victories, which naturally encouraged a ToM-centered interpretation. In the revised version, we removed that structure and replaced it with a **draw-based interval** rule. Under this revised environment, we also introduced the discussion of *strategic withdrawal*: in narrow regimes, agents may face situations where reliably satisfying the win condition is difficult, while reaching a draw remains feasible.
>
> Second, we revised the paper’s framing and interpretation to match the new environment. In §1, we removed the stronger claim that successful adaptation requires ToM, and instead reframed the benchmark as one for adaptation under latent objectives in adversarial play. In §5.2, we no longer explain narrow-regime difficulty mainly through the lack of ToM. Instead, the revised discussion emphasizes that under the draw-based rule, low win rates can coexist with high draw rates, meaning that agents may still display meaningful non-losing behavior even when they do not reliably convert to wins.
>
> We limit ToM as a viable future work in §6, and include the Hanabi challenge (Bard et al., 2020) as a related work in §6.
>
> **3. PPO Baseline**
>
> We agree that the original PPO baseline was too weak; thus, we revised it substantially to make the comparison more appropriate.
>
> The original version used a simple setup against the Random agent. In the revised manuscript, PPO has been redesigned to better match the benchmark setting (§4.4, Appendix F). Concretely, PPO is now trained under the same **2,000-game interaction budget** as the adaptive Minimax agent; the training protocol now uses **self-play** with a masked PPO formulation with valid moves given.
>
> We also conducted an evaluation of PPO's performance against Random and MCTS-100 under increasing training budgets of [200, 2000, 10000] to show the convergence speed in Figure 2, and reported the results of Adaptive Minimax against PPO-2000 in Table 1. It is shown that the PPO's performance has significantly increased compared to the previous version, especially against the Random agent, yet PPO remains weaker than Minimax in most environments (§5.2). Furthermore, the performance gain from 2000 games to 10000 games of training was relatively small (Figure 2).
>
> **4. Confidence Bounds in Table 1**
>
> We also agree that the confidence bounds in Table 1 can appear large. We clarified this in Table 1 and expanded to the scores for each specific environment in Tables 2 and 3 in the appendix.
>
> In the previous manuscript, Table 1 aggregated results in two ways: across environments sharing the same layout (top), and across environments sharing the same victory condition (bottom). Because performance varies substantially across victory conditions, the same-layout aggregation produced very large standard deviations. In the revised manuscript, we report the Win/Draw/Lose rate of adaptive minimax only aggregated across environments sharing the same victory condition, and per-environmental results are shown in Tables 2 and 3.

---

### Review · Reviewer_wa37 · 2026-03-23

**Summary Of Contributions:**

This paper presents the Expanded Othello AI Arena, a new benchmark for evaluating agents' ability to function in zero-sum dynamic environments. Notably, the benchmark extends the traditional Othello game by separating the board shape (denoted by L) and the conditions for winning (denoted by C). The paper presents a disc-ratio threshold K for parameterizing C. In a traditional Othello board, L is an 8x8 board and K > 1.0. In the Expanded Othello AI Arena, L and C are varied across several options producing 7 board options for L and 8 K-value options for C( K >1.0, K =0.8, K =0.6, K =0.4, K = 0.2, K < 0, K >1.0 10 turns limit, K < 0.0 10 turns limit)  arriving at a total of 56 unique environments. The  Expanded Othello AI Arena gives agents 2000 interactions per environment, and the agent is supposed to learn the rules over those 2000 interactions while competing against an opponent. The Arena is set up not to provide any rewards or signals until the game/competition is terminated, so the agent can only infer the rules for each environment from termination signals. The Arena allows the evaluation of how efficiently an agent can acquire skills in an adversarial setting.  The paper discusses in depth different aspects of the benchmark: including formalizing the adaptation task, the dynamics in the game, the environment (including the board layout, the various regimes and variants achieved by varying K), and the interaction protocol. The paper next presents the 56 unique environments, the variations of L and C (K) that generate them, and the evaluation constraints: 2000 interactions per environment, no transfer of knowledge between environments, and what prior knowledge is allowed. The paper provides a neuroevolutionary adaptive-Minimax architecture as a baseline and then has other baseline agents: a random agent, 3 Monte Carlo Tree Search (MCTS) agents, and PPO, and presents results from them. Future research directions and related works are also presented.


Key strengths of the paper include the robust, in-depth discussion of the environment, parameterizations, equations, winning condition variants, and agents' setup in both the main paper and the appendix.

**Additional Comments:**

On a minor note, the paper contains a few grammatical or formatting errors that need correction and would benefit from general proofreading by the authors. For example, for Table 1, it is recommended that the authors show the best performing values for each layout/winning condition in bold,  as well as provide a guide (e.g., arrows) alongside the table to show that lower values are better (MCTS-100 < MCTS-50 < MCTS-30 < PosNet < Random).

**Audience:**

Yes

**Audience Explanation:**

The findings of this paper would be of interest to researchers and audience members in the fields of Machine Learning Theory, Algorithmic Game Theory, Reinforcement Learning (RL).

**Broader Impact Concerns:**

The paper has no Broader Impact statement section.

Beyond the known concerns about training costs for machine learning projects, no broader, unique impact concern comes to mind.

**Claims And Evidence:**

Yes

**Claims Explanation:**

The paper is quite detailed in its setup and explanation of the Expanded Othello AI Arena, supported by motivation, equations, evaluations, and results.

**Requested Changes:**

1. The authors need to clarify the role of 'theory of mind (TOM)' as discussed in the paper. Section 1 states that successful adaptation in Othello demands functional integration of TOM, and TOM is stated as a requirement for favorable outcomes (contribution 2). However, in Section 5.2, while discussing why the Minimax agent fails in the narrow-interval regimes, the paper states that "...performance degradation primarily stems from the lack of a functional theory of mind". Given that the agent performs better in other configurations, is the paper's framing of TOM as a requirement for success in Othello, in general, wrong or too broad? Or does something about the narrow-interval regimes affect how TOM works?

2. Is the 5x5 window chosen because of the 2-cell obstacle padding around the input? Otherwise, it seems a 4x4 window still provides a 2-cell radius sufficient to perceive corners or edges from the critical squares.

---

> ### Author Response · Authors · 2026-04-03
> **Response to reviewer wa37**
>
> We sincerely thank the reviewer for the positive assessment and constructive feedback. In the revision, we clarified the role of Theory of Mind (ToM) by revising both the environment semantics and the paper’s framing, and we clarified the rationale for the 5×5 kernel choice.
>
> ***
>
> **1. Clarifying the role of Theory of Mind (ToM)**
>
> We agree that the original manuscript framed ToM too broadly and heavily. In the revision, we addressed this in two distinct ways. First, we modified the environment itself. Second, we changed the paper's description and interpretation accordingly.
>
> **A. Environmental Changes [§2.3.2]**
>
> The main revision was made in §2.3.2, where we changed the semantics of the narrow regimes. In the original version, narrow regimes explicitly included **opponent-forfeit-based** victories. This naturally encouraged a ToM-centered framing, as successful play could be described as strategically inducing the opponent to violate the hidden threshold.
>
> In the revised version, we replaced that structure with a draw-based interval rule. Violating the admissible interval no longer gives the other player a win; instead, it results in a **draw**. This makes the benchmark **less centered on opponent-inducement** and more centered on whether an agent can regulate its terminal occupancy under a latent objective.
>
> In the revised version, we also introduced the discussion of *strategic withdrawal*. This follows from the revised draw-based rule: in narrow regimes, an agent may face states where satisfying the winning condition is difficult, while reaching a draw remains feasible. In those cases, moving toward a draw can be a rational adaptive response. This contrasts with the original version, which instead emphasized opponent-forfeit inducement.
>
> **B. Revision of the Paper's Claim and Perspective**
>
> In the Introduction and Contributions, we removed the stronger claim that successful adaptation requires ToM, and instead described the benchmark as one for **adaptation under latent objectives in competitive zero-sum play**. This change was important because the earlier framing made it sound as though success in the Arena generally depended on opponent-intention reasoning, whereas the revised environment no longer supports such a strong interpretation.
>
> We also revised the interpretation in §5.2. In the original version, narrow-regime difficulty was explained mainly through the lack of ToM. In the revised manuscript, we no longer attribute the difficulty to ToM. Instead, we interpret the results in a way that is consistent with the revised rule structure: when violating the interval leads to a draw rather than an opponent win, narrow-regime adaptation should not be read only through win rates. Under this setting, low win rates can coexist with meaningful non-losing behavior, and the revised text therefore emphasizes that high draw rates often accompany narrow-regime outcomes. This suggests that agents are often able to avoid clearly losing terminal outcomes even when they do not reliably achieve wins, which is consistent with the revised discussion of strategic withdrawal.
>
> This change also affects the overall perspective of the paper. Rather than presenting the benchmark as one that directly demonstrates the necessity of ToM, the revised manuscript presents it more carefully as a benchmark for adaptation under latent objectives in adversarial play, while acknowledging that richer forms of opponent modeling may become more important in future variants.
>
> Finally, in Future Directions, we further weakened the ToM claim by discussing it only as a possible extension rather than a requirement.
>
> **2. Clarification of the $5\times5$ Kernel**
>
> We agree that the original explanation was unclear. We clarified that the main reason for choosing a 5×5 kernel instead of a 4×4 kernel is not simply the local radius, but the need for an odd-sized receptive field with a unique center cell for square-centered positional evaluation (Appendix C.1).
>
> We also corrected minor grammatical and presentation issues during revision.

---

> > ### Comment · Reviewer_wa37 · 2026-04-28
> >
> > I thank the authors for addressing my comments and concerns. The paper is in a better position now.

---

### Comment · Action_Editor_dxHJ · 2026-03-10
**[Reviewer action required] Review due**

Dear reviewers, if you haven't already done so, please submit your review for this paper.
Thank you if you have already submitted.

Thanks for your work and engagement,
  AE

---

### Decision · Action_Editor_dxHJ · 2026-05-10

**Recommendation:** Accept as is

**Audience:**

Yes

**Audience Explanation:**

The benchmark setting is interesting and timely, and is relevant for TMLR's audience interested in meta-learning, information-gathering in an adversavial decision-making setting, and learning under constrained interaction budget.

**Claims And Evidence:**

Yes

**Claims Explanation:**

The paper introduces a benchmark, based on the game Othello, where an agent is evaluated in its capability to rapidly adap to a non-fully-observable environment in an adversarial (zero-sum) setting using a constrained interaction budget. To do so, the authors create 56 variations / extensions of Othello and measure agents' ability to adapt to unseen versions over 2000 games. In addition to the benchmark the authors supply 3 baseline agents: a minimax agent that uses learned priors, a model-free PPO agent, and a simulator-privileged comparison using MCTS (that provides interesting additional info, but has sampling-based access to privileged information).

All reviewers agree that the benchmark is an interesting and novel addition, and that the results are accurate, clear, and convincing. The intial version of the manuscript made some claims regarding the potential of the benchmark suite to study theory-of-mind modeling, which was criticized by two reviewers as an interpretation (and maybe aspiration for future analysis) rather than a result. Another criticism was that 2000 games may be a relatively long interaction phase for some of the claims. The authors revised the manuscript to clarify both issuse (including the extension to analysis over the course of 2000 games) and address other minor criticism. After the rebuttal and author-reviewer discussion phase, all reviewers agree that all claims in the manuscript are supported by accurate, convincing, and clear evidence; and I agree with that assessment.